# The structure of basal body inner junctions from *Tetrahymena* revealed by electron cryo-tomography

Sam Li [1✉], Jose-Jesus Fernandez [2], Marisa D Ruehle[3], Rachel A Howard-Till[4], Amy Fabritius[4], Chad G Pearson[3], David A Agard [1,5✉] & Mark E Winey [4✉]

## Abstract

**The cilium is a microtubule-based eukaryotic organelle critical for many cellular functions. Its assembly initiates at a basal body and continues as an axoneme that projects out of the cell to form a functional cilium. This assembly process is tightly regulated. However, our knowledge of the molecular architecture and the mechanism of assembly is limited. By applying cryo-electron tomography, we obtained structures of the inner junction in three regions of the cilium from *Tetrahymena*: the proximal, the central core of the basal body, and the axoneme. We identified several protein components in the basal body. While a few proteins are distributed throughout the entire length of the organelle, many are restricted to specific regions, forming intricate local interaction networks in the inner junction and bolstering local structural stability. By examining the inner junction in a POC1 knockout mutant, we found the triplet microtubule was destabilized, resulting in a defective structure. Surprisingly, several axoneme-specific components were found to "infiltrate" into the mutant basal body. Our findings provide molecular insight into cilium assembly at the inner junctions, underscoring its precise spatial regulation.**

**Keywords** Basal Body; Centriole; Cilium; Assembly; Electron Cryo-tomography
**Subject Categories** Cell Adhesion, Polarity & Cytoskeleton; Structural Biology

## Introduction

The cilium or flagellum is a hair-like organelle that fulfills many cellular functions, ranging from cell motility to cellular signaling. The cilium assembly initiates when a barrel-shaped basal body (BB) docks underneath the cell membrane and serves as a template for cilium formation. This is followed by building an axoneme at the distal end of the BB, which elongates and projects from the cell surface until it reaches a certain length. Hundreds of proteins are directly involved in the cilium assembly process, which is highly regulated. Mutation of the ciliary components or dysregulation of the assembly process leads to defective cilia with complex phenotypes. These are manifested in humans as ciliopathies, a diverse spectrum of diseases such as microcephaly and primary ciliary dyskinesia (Reiter and Leroux, 2017; Mill et al, 2023).

Cilium assembly is an evolutionarily conserved process (Carvalho-Santos et al, 2010, 2011; Cavalier-Smith, 2014; Jana, 2021). Previous studies show that the centriole and BB exhibit longitudinal ultrastructural variations that can be demarcated into several regions, namely, the proximal, the central core, and the distal regions (Allen, 1969; Geimer and Melkonian, 2004; Greenan et al, 2018; Li et al, 2019; LeGuennec et al, 2021; Zhang et al, 2024; Laporte et al, 2024; Ruehle et al, 2024). In *Tetrahymena*, a new BB first emerges orthogonal to the proximal end of the old BB. This is marked by a tubular hub at the center of the BB with nine spokes radially emanating from the hub. Together, they form a cartwheel scaffold that defines the organelle's overall ninefold symmetry (Breugel et al, 2011; Kitagawa et al, 2011; Noga et al, 2022). At the tip of each spoke, a pinhead structure connects the cartwheel to the triplet microtubule (TMT). The TMT comprises a complete A-tubule with 13 protofilaments (pfs) and partial B- and C-tubules that share pfs with the neighboring tubules. Each TMT is linked to its neighbor TMT by a structure called an A–C linker. This proximal region of the BB longitudinally spans ~150 nm (Fig. 1A). In the central core region, the cartwheel terminates and is replaced by an inner scaffold (Fig. 1B), a conserved sheath-like structure at the inner circumference of the BB barrel. The inner scaffold is critical for overall BB structural stability and resistance to the external force exerted onto the BB during cilium beating (Guennec et al, 2020; Ruehle et al, 2024). The central core region spans about 300 nm longitudinally and is followed by the distal region that longitudinally spans ~110 nm. Here, the BB anchors to the plasma membrane and the TMT becomes the microtubule doublet (DMT), where the C-tubules terminate while the A- and B-tubules continue extension. At the distal end of the BB, a specialized compartment called the transition zone marks the transition of the BB to the ciliary axoneme.

[1]Department of Biochemistry and Biophysics, University of California San Francisco, San Francisco, CA 94158, USA. [2]Nanomaterials and Nanotechnology Research Center (CINN-CSIC), Health Research Institute of Asturias (ISPA), 33011 Oviedo, Spain. [3]Department of Cell and Developmental Biology, University of Colorado Anschutz Medical Campus, Aurora, CO 80045, USA. [4]Department of Molecular and Cellular Biology, University of California Davis, Davis, CA 95616, USA. [5]Chan Zuckerberg Institute for Advanced Biological Imaging, Redwood City, CA, USA. ✉E-mail: samli@msg.ucsf.edu; david.agard@czii.org; mwiney@ucdavis.edu

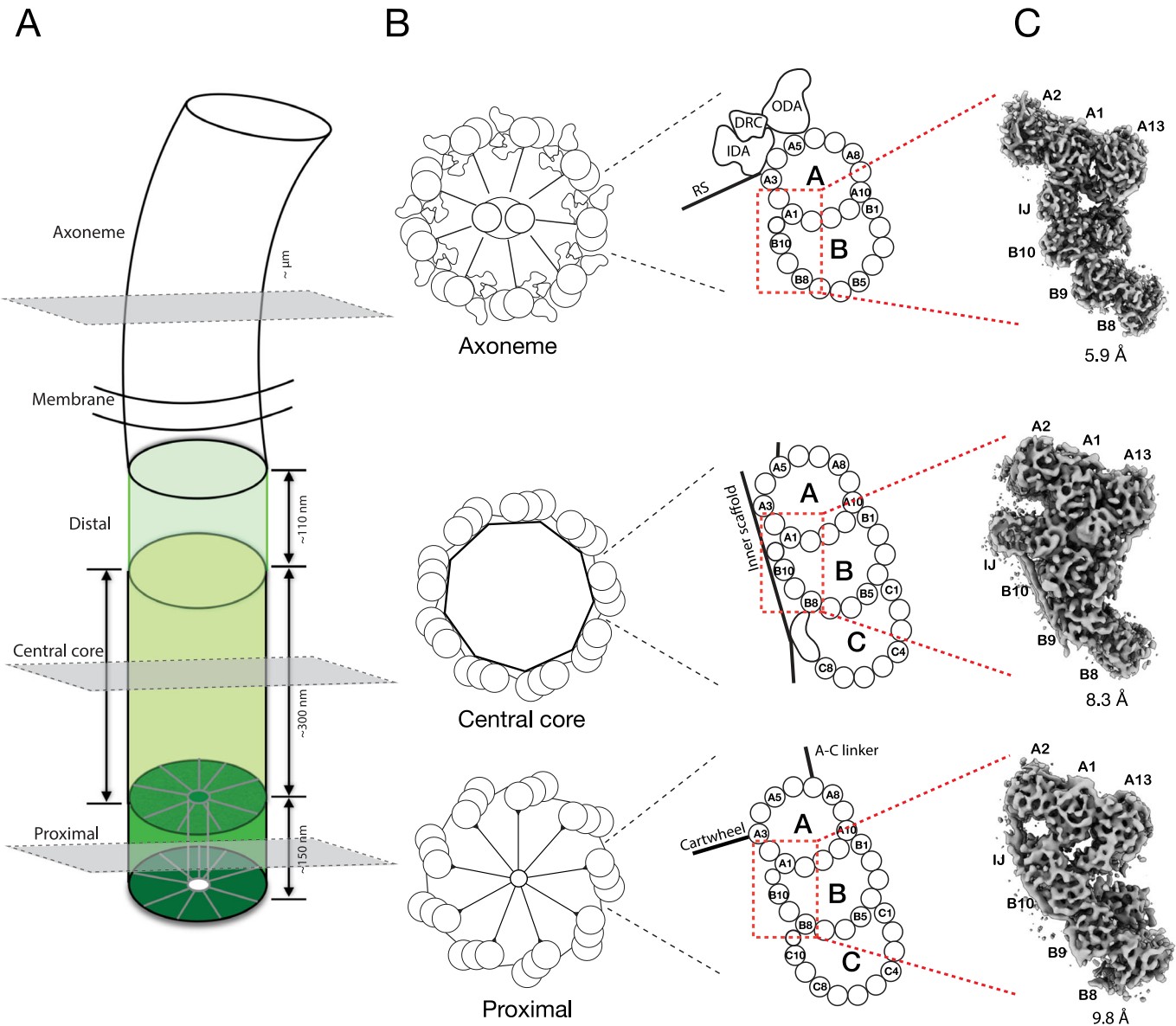

**Figure 1. Electron cryo-tomography structures of cilium inner junctions.**

(A) A schematic diagram of a cilium in *Tetrahymena*, including BB and axoneme. The three regions in the BB, the proximal, central core, and distal, are highlighted in different green colors. Their approximate longitudinal spans are indicated. The three gray-colored cross-sections indicate the location of the structures presented in this work. (B) Left, schematic diagrams of the cross-section of the proximal, the central core region of the BB, and the axoneme. Right, representation of the triplet MT or doublet MT from the three regions. Distinct structures attached to the microtubule wall in each region, such as the cartwheel and the A–C linker in the proximal region, the inner scaffold in the central core region, and the Dynein complexes (ODA outer Dynein arm, IDA inner Dynein arm, DRC Dynein regulatory complex) and the radial spokes (RS) in the axoneme, are indicated. The red dashed line boxes highlight the A–B inner junctions (IJ). (C) Three representative subtomograms-averaged structures in 16-nm periodicity are presented in this work. From bottom to top are the proximal, the central core region of the BB, and the axoneme, as indicated in the cartoons in (B).

The morphological differences in BB ultrastructure indicate variation in its composition along the length of the BB and axoneme. Immuno-EM, super-resolution light microscopy and, in particular, recent advances in Ultrastructure Expansion Microscopy (UExM) have greatly enriched our knowledge of the molecular composition and organization of the BB (Pearson et al, 2009; Hamel et al, 2017; Guennec et al, 2020; Tian et al, 2021; Steib et al, 2020; Arslanhan et al, 2023; Laporte et al, 2024). These studies show that the different BB regions have unique sets of proteins.

These composition specificities imply the existence of a spatio-temporal control mechanism that governs the BB assembly. However, the molecular architecture of the BB, the precise boundaries of the composition changes, and the mechanism governing the BB assembly need to be better understood.

The inner and outer junctions, where the B- or C-tubule joins to the neighboring A- or B-tubule, respectively, are unique structures in the TMT and DMT. The inner junction faces the luminal side of the BB, whereas the outer junction faces the exterior of the BB

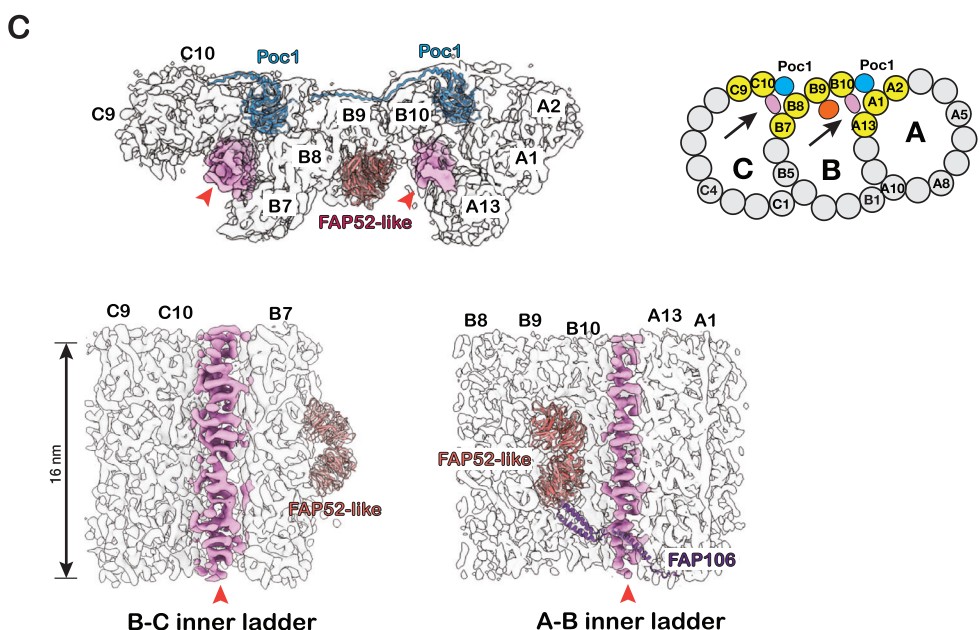

◀  **Figure 2.  Structure of the inner junctions in the proximal region of the BB.**

(**A**) Two orthogonal views of the A–B inner junction structure with 16-nm repeat. A FAP52-like MIP, FAP106, and Poc1 are colored in red, purple, and blue, respectively. A schematic diagram of TMT is on the right. The pfs shown in the structure are highlighted in yellow. A dash line indicates the cutting plane and an arrow indicates the direction of view. (**B**) Comparison of the A–B inner junctions in the BB's proximal and central core regions. The proximal region structures are on the left, and the central core region structures are on the right. The MIP models and the tubulins (α-tubulins in green, β tubulin in light blue) are fit into the density maps in gray. Note a FAP52-like MIP binds to the B-tubule at a different location in the proximal region compared to the FAP52 in the central core region. (**C**) Upper: a composite map showing the A–B and B–C inner junctions in the proximal region. The A–B and B–C inner ladders crosslinking pfs A13-B10 and B7-C10 are highlighted in light pink and indicated by red arrowhead. A schematic diagram of TMT is on the right. The pfs shown in the structure are highlighted in yellow. The arrows indicate the directions of view in the bottom panels. Bottom: longitudinal cross-section views of the A–B and B–C inner junctions.

(Fig. 1B). Recent studies show that the axonemal inner junction is a protein interaction hub in motile cilia, playing a critical role in withstanding substantial force and stress during cilia beating (Owa et al, 2019; Khalifa et al, 2020). Therefore, it is not surprising that many microtubule inner proteins (MIPs) identified in the axoneme inner junction, such as FAP20, PACRG, FAP52/WDR16, FAP106/ENKUR, FAP45, and FAP210, are evolutionarily conserved across many species (Gui et al, 2021; Kubo et al, 2023). Mutations in many of their human orthologs cause ciliopathies (Zhou et al, 2023; Walton et al, 2022; Gui et al, 2021; Ma et al, 2019; Leung et al, 2023; Sigg et al, 2017; Kubo et al, 2023). However, in contrast to our rich knowledge about the axoneme inner junction structures, little is known about the inner junction in the BB. This limits our understanding of its molecular composition and the mechanistic details of its assembly.

Recently, we identified Poc1 as a key component in the inner junctions critical for BB structure stability and integrity (Ruehle et al, 2024). Here, using cryogenic electron tomography (cryoET) and subtomogram averaging, we extended the study of basal bodies and axonemes isolated from wild-type and mutant strains of a unicellular model organism, *Tetrahymena thermophila*. Based on the inner junction structures in subnanometer resolution, representing three locations of the cilium that cover both BB and axoneme (Figs. 1C and EV1), we identified several components that are universal along the cilium and proteins that are unique to specific regions. This spatial specificity at the inner junction underpins a mechanism that might be applied to the entire cilium assembly during the organelle's biogenesis.

## Results

### Structure of the inner junction at the BB proximal region

We isolated BBs and flagella from *Tetrahymena*, respectively. This was followed by collecting tomographic tilt series and reconstructing 3D tomograms of BBs and flagella. First, we analyzed the TMT inner junction from the BB's most proximal ~150 nm region by subtomogram averaging. At 9.8 Å resolution (Figs. 2A and EV1A,B; Movie EV1), the fold of tubulins and many MIPs could be resolved in the structure. This facilitated identifying and fitting their corresponding atomic models into the density maps. In addition to the Poc1 that has recently been identified in the A–B and B–C inner junctions (Ruehle et al, 2024), we identified a FAP52-like protein and FAP106 in the A–B inner junction in the proximal region (Fig. EV2A). This suggests that all three components, FAP52-like, FAP106, and Poc1, are recruited to the TMT at the beginning of BB assembly. Like the axoneme, the

FAP52-like protein and FAP106 have a longitudinal periodicity of 16 nm (Fig. 2A). A comparison of the binding of FAP52 to the microtubule (MT) backbone in the proximal region to the central core region reveals marked differences (Fig. 2B). Surprisingly, this FAP52-like protein in the proximal region binds predominantly to pf B9 (Fig. 2B). This is in contrast to the core region, where FAP52 ($FAP52_{core}$) is at the lateral interface between pf B9 and B10, interacting with both pf, which resembles the structure in axoneme DMT (Ma et al, 2019; Khalifa et al, 2020).

In the proximal region, in addition to Poc1, FAP52-like protein, and FAP106, we found two previously unidentified structures in the A–B and B–C inner junctions. They connect the A- and B-tubules or B- and C-tubules by crosslinking pf B10 to A13 or C10 to B7, respectively (Figs. 2C and EV2B). Longitudinally, both structures resemble ladders composed of a stack of rungs. Thus, we named them the A–B inner ladder and B–C inner ladder. Both are unique to the proximal region. Since the A–B inner ladder occupies the luminal side of pf B10, this A–B inner ladder and the $FAP52_{core}$ at its canonical position between pf B09 and B10 are mutually exclusive.

Interestingly, these A–B and B–C inner ladders remain present in two inner junction locations in the POC1 knockout mutant (poc1Δ) (Fig. EV2C), suggesting they have overlapping functions with Poc1 to crosslink the inner junctions and stabilize the TMT in the proximal region. We cannot confirm if these two structures are composed of an identical MIP at our current resolution. It will be interesting to identify these components and study their function in the future.

### The transition from the proximal to the central core region of BB

The finding that a FAP52-like protein binds at different locations in the proximal and the core regions implies that the molecule will shift its location near the end of the proximal region before reaching the core region. To investigate this shift in more detail and to identify any additional changes associated with this shift, we applied 3D classification on the subtomograms dataset from the BB proximal region, focusing on the A–B inner junction. The result shows two groups where FAP52 is at different locations (Figs. 3A and EV3A). A predominant group (Class 1, 6030 subtomograms, 76.8%) shows the FAP52-like protein binding at pf B9, representing the proximal region structure. In a minor group (Class 2, 1218 subtomograms, 15.5%), the FAP52 binds to pf B9/B10, a "canonical" binding location by FAP52 both in the central core region of BB and in the axoneme (Figs. 3A and EV3A). We backtracked their location in the BB and generated a histogram representing the longitudinal distribution of these two groups

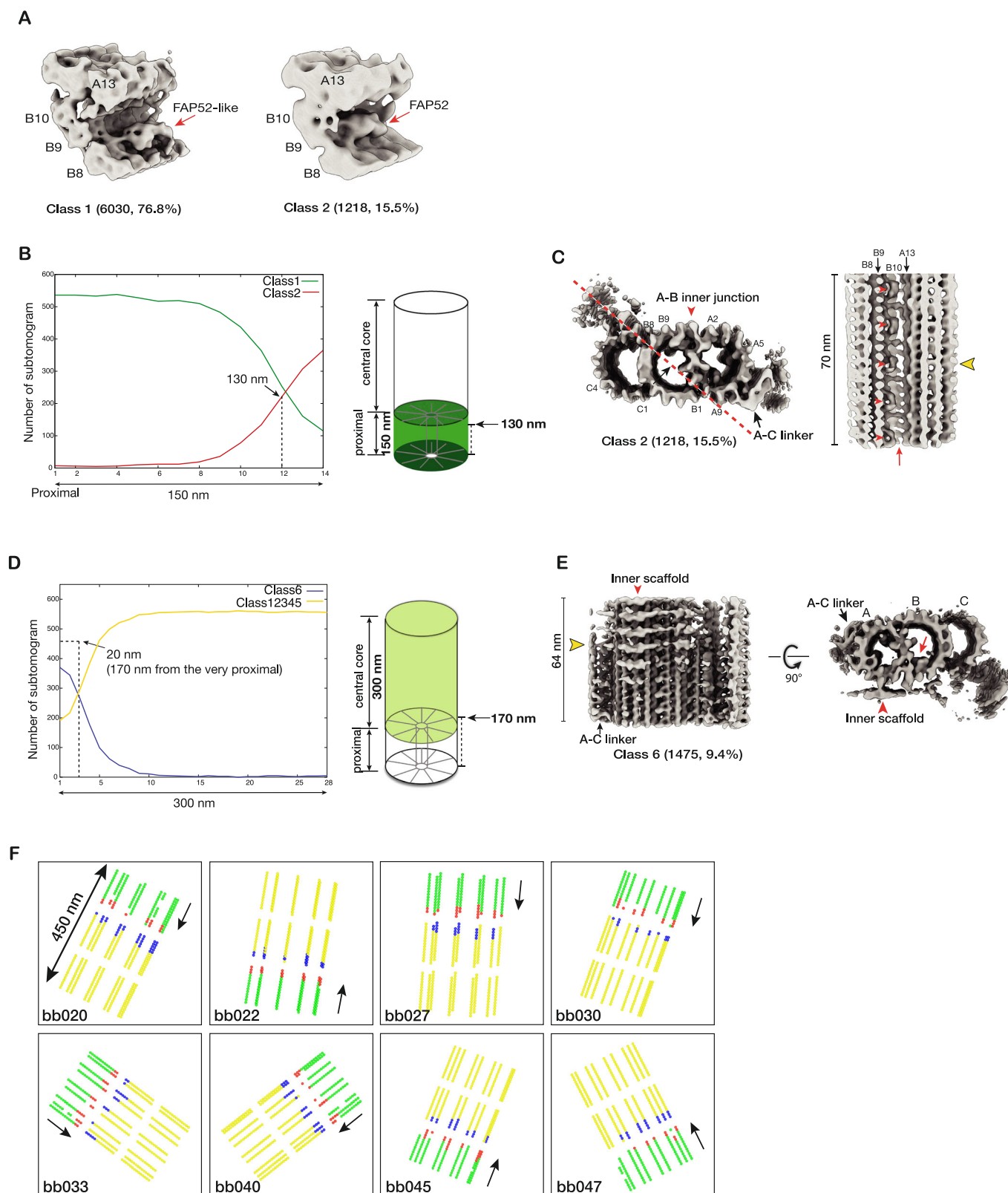

**A**

Class 1 (6030, 76.8%)

Class 2 (1218, 15.5%)

**B**

130 nm

Proximal 150 nm

**C**

Class 2 (1218, 15.5%)

**D**

20 nm
(170 nm from the very proximal)

300 nm

**E**

Class 6 (1475, 9.4%)

**F**

bb020  bb022  bb027  bb030

bb033  bb040  bb045  bb047

(Fig. 3B). The distribution histogram shows that the transition occurs approximately at a longitudinal position of 130 nm. This minor group likely represents the structure near the distal end of the proximal region, where the FAP52-like protein shifts from pf

B9 to the FAP52 in canonical position at pf B9/B10. To visualize this shift, we calculated the average of this minor group in a larger volume, longitudinally spanning 80 nm, including additional structure features towards the proximal end, where it shows the

◀ **Figure 3. Structural changes at the transition from the proximal to the central core region of the BB.**

(A) Focused classification on the subtomograms from the proximal region identifies two structures where a FAP52-like protein binds at different locations (indicated by red arrows). (B) The longitudinal distribution of the subtomograms from the proximal region. The 150 nm long proximal region is divided into 14 bins. Based on the two classes identified in (A), for each class, the number of subtomograms (y axis) found in each bin is plotted, showing their longitudinal distribution along the length of the proximal region (x axis). Right: A schematic diagram indicates the weighted average longitudinal location for Class 2 is at 130 nm. The weighted average length is $L = \sum j * Nj/\sum Nj$ (Nj: number of subtomograms found in bin j). (C) Left: The Class 2 average in a cross-section view. A red arrowhead indicates the A–B inner junction. A red dashed line and a black arrow indicate the cross-section and viewing direction of the structure on the right. Right: The averaged structure from Class 2, longitudinally spanning 70 nm. A yellow arrowhead indicates the location where the FAP52-like MIP (red arrowheads) shifting binding site. The shift coincides with the termination of the unidentified ladder-like protein in the A–B inner junction (red arrow). (D) Similar to (B), the longitudinal distribution of the subtomograms from the central core region. The 300 nm long central core region is divided into 28 bins. Based on the classification result shown in Fig. EV3C, the longitudinal distribution of subtomograms in Class 6 and the other five classes are plotted. The weighted average longitudinal position of Class 6 is at 170 nm from the very proximal end of the BB. This is illustrated in a schematic diagram on the right. (E) The averaged structure in Class 6 (Fig. EV3C) shows the changes of TMT transitioning from the proximal to the central core region. A red arrowhead indicates the inner scaffold. A red arrow indicates FAP52. A yellow arrowhead indicates the termination of the A–C linker and the emergence of the inner scaffold. (F) Mapping the distribution of subsets of subtomograms in the BB. Eight representative BBs are shown. Each dotted line represents a TMT, longitudinally spanning 450 nm from the proximal to the core region (an arrow indicates the polarity). The 450 nm longitudinal span is divided into 3 segments, each 150 nm long. The green dots represent the Class 1 subset in (A), an inner junction structure in the proximal region. The red dots represent subtomograms in Class 2 in (A) and EV3A, where a FAP52-like protein shift from pf B9 to FAP52 at the canonical position between pf B9 and B10. The blue dots represent Class 6 in (E) and EV3C, where the A–C linker recedes, and the inner scaffold emerges, marking the transition from the proximal to the central core region. The yellow dots represent the structure in the core region, where the A–C linker is absent, and the inner scaffold is fully assembled in the TMT.

FAP52-like protein binding to pf B9 (Figs. 3C and EV3B; Movie EV2). Thus, this extended volume confirms the change of FAP52 taking place at ~130 nm.

Interestingly, the shift of FAP52-like MIP from its pf B9 position to the FAP52 at canonical pf B9/B10 position is concomitant with the termination of the A–B inner ladder at the inner junction (Fig. 3C; Movie EV2). Likely, the termination of this A–B inner ladder at 130 nm longitudinal position makes space for FAP52. Furthermore, the A–C linker, a characteristic feature of the proximal region, remains in the structure at this longitudinal position. In contrast, the inner scaffold, a structure unique to the BB central core region, is absent (Fig. 3C). This indicates that, at this point, the central core region has yet to be established when the FAP52-like protein shifts the location.

To locate and visualize the transition from the proximal to the central core region, we applied 3D classification on the subtomograms dataset from the BB central core region. The classification identified a minor subset (1475 subtomograms, 9.4%) showing both the A–C linker and the inner scaffold (Fig. EV3C). A longitudinal distribution plot shows that, in contrast to other subsets with subtomograms evenly distributed throughout the central core region, this subset is concentrated at the beginning of the central core region, centered at 170 nm longitudinal position (Fig. 3D). A 3D average of this subset shows the A–C linker's wane and the inner scaffold's concomitant emergence (Fig. 3E; Movie EV3). This likely marks the transition from the proximal to the central core region of the BB.

In summary, our structural analysis of the BB's most proximal 150 nm stretch shows a series of sequential events in the inner junctions. First, we found a FAP52-like protein, FAP106, and Poc1 in the proximal region of the BB. Surprisingly, this FAP52-like protein binds at pf B9, while two unidentified components forming ladder-like structures crosslink A–B and B–C inner junctions, respectively. Second, at about 130 nm from the proximal end, the inner ladders terminate. This coincides with the FAP52-like protein shifting from pf B9 to FAP52 in its "canonical" position at pf B9/B10. Third, at ~170 nm, the A–C linker terminates and the inner scaffold emerges, marking the transition from the proximal to the central core region. All these changes occur in well-defined longitudinal positions and are displayed sequentially,

demonstrating a tight spatial regulation for the BB MIPs. We mapped the location of these changes in the BB and illustrated them in eight representative BBs in Fig. 3F.

## Structure of the inner junction in the core region and comparison to the flagellar axoneme

In *Tetrahymena*, the central core region of BB spans longitudinally about 300 nm, from 150 nm to 450 nm (Fig. 1). The A–B inner junction in this region has important structural roles. First, it crosslinks the A- and B-tubule in the luminal side of the BB, stabilizing the TMT. Second, it connects the TMT to the inner scaffold, a unique structure in the core region critical for the overall BB stability and structural cohesion (Guennec et al, 2020; Ruehle et al, 2024). Poc1, a WD40 family protein, occupies the A–B inner junction, providing an anchor that connects the inner scaffold to the TMT. In the axonemal DMT, alternating FAP20 and PACRG filaments fill the A–B inner junction (Ma et al, 2019; Gui et al, 2021; Khalifa et al, 2020; Dymek et al, 2019). Based on the previously solved axoneme structures (Ma et al, 2019; Kubo et al, 2023), we built atomic models into the subnanometer density maps from the BB core and axoneme regions (Figs. 4A,B and EV4A; Movies EV4 and EV5). This allowed us to identify several axoneme MIPs in the A–B inner junction of the BB core region. These include FAP52, FAP106, IJ34, FAP45, and FAP210. These MIPs are in the same locations as the axoneme DMT, displaying the same longitudinal periodicity.

In addition to these MIPs in both BBs and axonemes, we observed a MIP unique to the BB core region (Fig. 4C,D). It crosslinks the A- and B-tubule by binding to pfs A13 and B10 with an 8 nm periodicity. At 8.3 Å resolution (Fig. EV1D), the map shows an overall double-layer arch-shaped topology (Fig. EV4B). The top layer comprises at least seven α-helices, whereas the bottom layer is continuous, likely a β-sheet. This tertiary arrangement suggests a leucine-rich repeat (LRR) motif. Searching the BB proteome database identified seven proteins containing LRR motifs (Kilburn et al, 2007). Based on the AI-based structure prediction (Jumper et al, 2021), one of the proteins' (UniProt Q22N53) C-terminal LRR motif fits well into the density map (Figs. 4C and EV4C; Movie EV6). In the model, the LRR is an arch

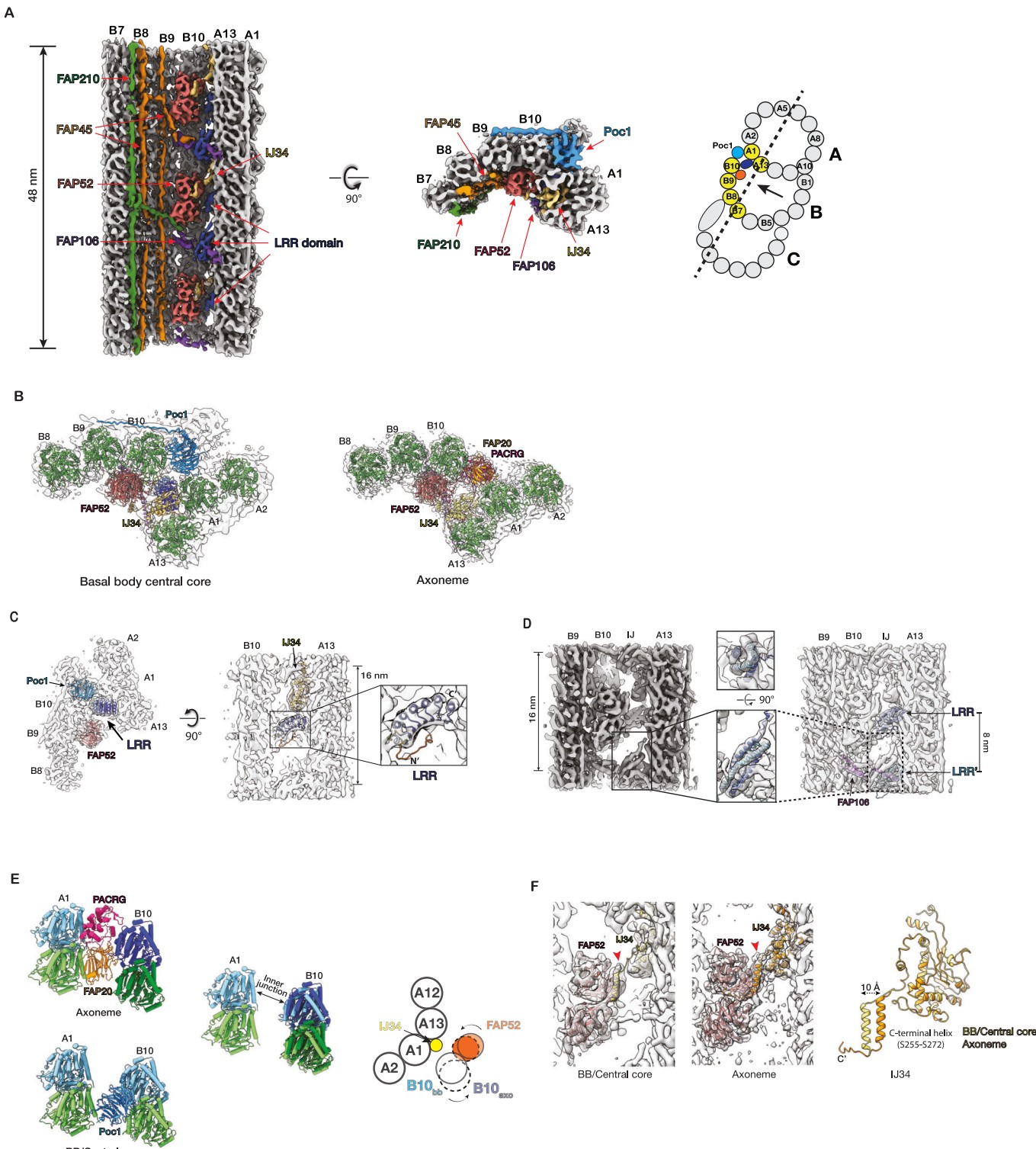

connecting the A- and B-tubule at the inner junction. Specifically, the N-terminal end of the LRR motif potentially contacts an α-tubulin from pf B10, while at the opposite side of the arch, the C-terminus of the LRR motif binds at a four-tubulin corner formed by pf A13 and A1. In addition, the LRR also makes potential contact with Poc1 and IJ34 (Fig. 4C; Movie EV6). The N-terminal

portion of this LRR protein is predicted as two α-helices forming an anti-parallel hairpin, albeit with low confidence scores (pLDDT) (Fig. EV4C). Interestingly, in the averaged density map we observed a right-handed 4-helix bundle between two neighboring LRR-containing proteins (Fig. 4D). This 4-helix bundle has 16-nm longitudinal periodicity and makes contact with FAP106. An

**Figure 4. Structure of the inner junctions in the central core region of the BB.**

(A) Two orthogonal views of a 48-nm repeat structure at the A–B inner junction in the BB central core region. The MIPs are highlighted in different colors as indicated. On the right is a schematic diagram of the TMT where the pfs shown in the structure are highlighted in yellow. A dash line indicates the cutting plane and an arrow indicates the viewing direction of the structure on the left. (B) Comparing the A–B inner junction in a 16-nm repeat from the BB central core region to the axoneme. The MIP models and the tubulins (represented by α-tubulins in green) are fit into the density maps in gray. (C) A MIP with a leucine-rich repeat (LRR) motif in the A–B inner junction. An LRR model predicted by AlphaFold2 from the protein (UniProt: Q22N53) fits into the density map. The LRR motif makes potential interactions with pfs B10, A13, and A1 of the TMT wall, Poc1, and IJ34. (D) An AlphaFold 3 predicted model of a right-handed 4-helix bundle fits into the density map. It is formed by two NTDs from the neighboring LRR-containing proteins as an anti-parallel dimer. Each monomer is colored in either dark or light blue. (E) Comparing the inner junctions between the central core region and the axoneme. Left: the models of the inner junction in the central core of BB and the axoneme. Center: superposition of the two models using pf A1 as a reference. Pf B10 (α/β tubulin) in the BB central core are in light green and blue. Pf B10 (α/β tubulin) in the axoneme are in dark green and blue. Right: schematic illustration of the change. The solid circles represent the pf B10 and FAP52 from the BB central core region. The dashed circles represent the pf B10 and FAP52 in the axoneme. The two curved arrows indicate the movement of pf B10 and FAP52 from the central core to the axoneme. (F) Comparing the structures of IJ34 in the central core region and the axoneme. The left and center show that IJ34 and FAP52 fit into the density maps. Right: superposition of the two IJ34 using their main domains as a reference. Their C-terminal helices are 10 Å apart.

AlphaFold 3 prediction (Abramson et al, 2024) shows that two copies of the α-helical hairpin could dimerize to form a 4-helix bundle (Fig. EV4D,E). This is further supported by fitting this predicted 4-helix bundle into the density map (Fig. 4D), showing an overall good agreement between the prediction and the experimental observation. Given the location of this LRR-motif MIP in the BB and its interactions with neighbors, it likely plays a critical role in stabilizing the inner junction in the central core region. However, the confirmation of its identity and its function in BB assembly and maintenance have to await future studies.

While Poc1 and the LRR-motif MIP occupy the inner junction in the BB central core region, the equivalent position in the axoneme is occupied by PACRG and FAP20 (Fig. 4B). This composition difference might lead to local structural alteration. A comparison of the microtubule arrangement in these two regions shows notable variation. Using the pf A1 as a reference, the relative position and the orientation of pf B10 change from the BB inner core region to the axoneme (Fig. 4E; Movie EV7). Compared to the BB core region, in the axoneme, pf B10 has shifted further away from the A-tubule (~6 Å), increasing the gap between pf A1 and pf B10 and enabling it to accommodate a PACRG and FAP20 protofilament. Meanwhile, pf B10 rotates counter-clockwise about 6 degrees (cartoon in Fig. 4E, viewed from the MT plus end). This shift and rotation of pf B10 brings its associated FAP52 closer to the A-tubule (cartoon in Fig. 4E). Interestingly, these structural changes in the MT lattice and FAP52 can be accommodated by IJ34, one of the inner junction MIPs present in both the BB and axoneme in *Tetrahymena* (Fig. 4F). The IJ34's C-terminal helix (S255-S272) can transverse about 10 Å and remain tethered to FAP52 in both structures, while its main globular domain anchors to the A-tubule (Fig. 4F). Likely, this is enabled by a flexible linker in IJ34 connecting the main globular domain to the C-terminal helix tail.

In addition to the structure change on MIPs, we also observed changes in the MT lattice. The lateral inter-protofilament curvature at pf B9/B10 is markedly higher in the core region (31.6°) compared to the proximal (20.1°) or the axoneme region (23.3°) (Fig. EV4F; Table 1), demonstrating the elasticity of the MT lattice that is capable of forming irregular and highly variable lateral curvatures. This is consistent with previous studies on different forms of chemically treated axoneme structures (Ichikawa et al, 2019), as well as a study on in vitro assembled MTs revealing its highly flexible and polymorphic lateral inter-pf interactions (Debs et al, 2020).

**Table 1. Measurement of inter-protofilament angle (Δφ, in degree).**

|  | Proximal | Central core | Axoneme | *POC1KO/* Axoneme-like subset (Class 3) | Ichikawa et al (Ichikawa et al, 2017) |
|---|---|---|---|---|---|
| A2/A1 | 23.6 | 24.8 | 26.6 | 24.6 | 26.1 |
| A1/A13 | 36.7 | 35.9 | 36.3 | 35.8 | 36.3 |
| B10/B9 | 20.1 | 31.6 | 23.3 | 27.6 | 24.5 |
| B9/B8 | 24.2 | 23.6 | 22.3 | 23.3 | 22.8 |

The inter-protofilament angles are determined by measuring the angle between Cα atoms of the same pair of residues in neighboring protofilaments.

## Incorporation of a set of axonemal proteins in the BB inner junction in the *POC1* knockout mutants

Recently, we identified Poc1 as a critical component at the inner junctions of TMT (Ruehle et al, 2024). In the proximal region, Poc1 is localized at both the A–B and B–C inner junctions. In the central core region, Poc1 is in the A–B inner junction, providing one of the anchoring sites for the inner scaffold. The study of poc1Δ mutation shows that the protein plays essential roles in BB stability and resisting external force (Junker et al, 2022; Ruehle et al, 2024). Here, we further characterize the poc1Δ defects in the BBs. Consistent with the previous observations, in poc1Δ, the BB is partially disintegrated, where the B-tubule is partially detached from the A-tubule. However, occasionally, we found that the B-tubule remained attached to the A-tubule at various longitudinal locations (Fig. 5A). To investigate this further, we classified the subtomograms from poc1Δ TMT, focusing on the A–B inner junction in the central core region. The classification identified three subsets showing distinct features in the inner junction (Fig. 5B). In the first subset (Class 1, 32.5% of the total population), the B-tubule is detached from the A-tubule, resulting in a gap at the inner junction between the A- and B-tubules. This is consistent with our recent observation that Poc1 is critical for BB stability (Ruehle et al, 2024). In the second subset (Class 2, 20.3%), the inner junction remains sealed, albeit without Poc1 (Fig. 5B). Further refinement of this subset shows that, despite the absence of Poc1 in the inner junction, many MIPs remain, including IJ34, the LRR-motif MIP, FAP106, and FAP52 (Fig. 5B,F). It is noticeable that in this second subset, while the LRR-motif MIP remains bound to pf A1/A13 in the A-tubule, it is partially detached from the B-tubule (Fig. 5F),

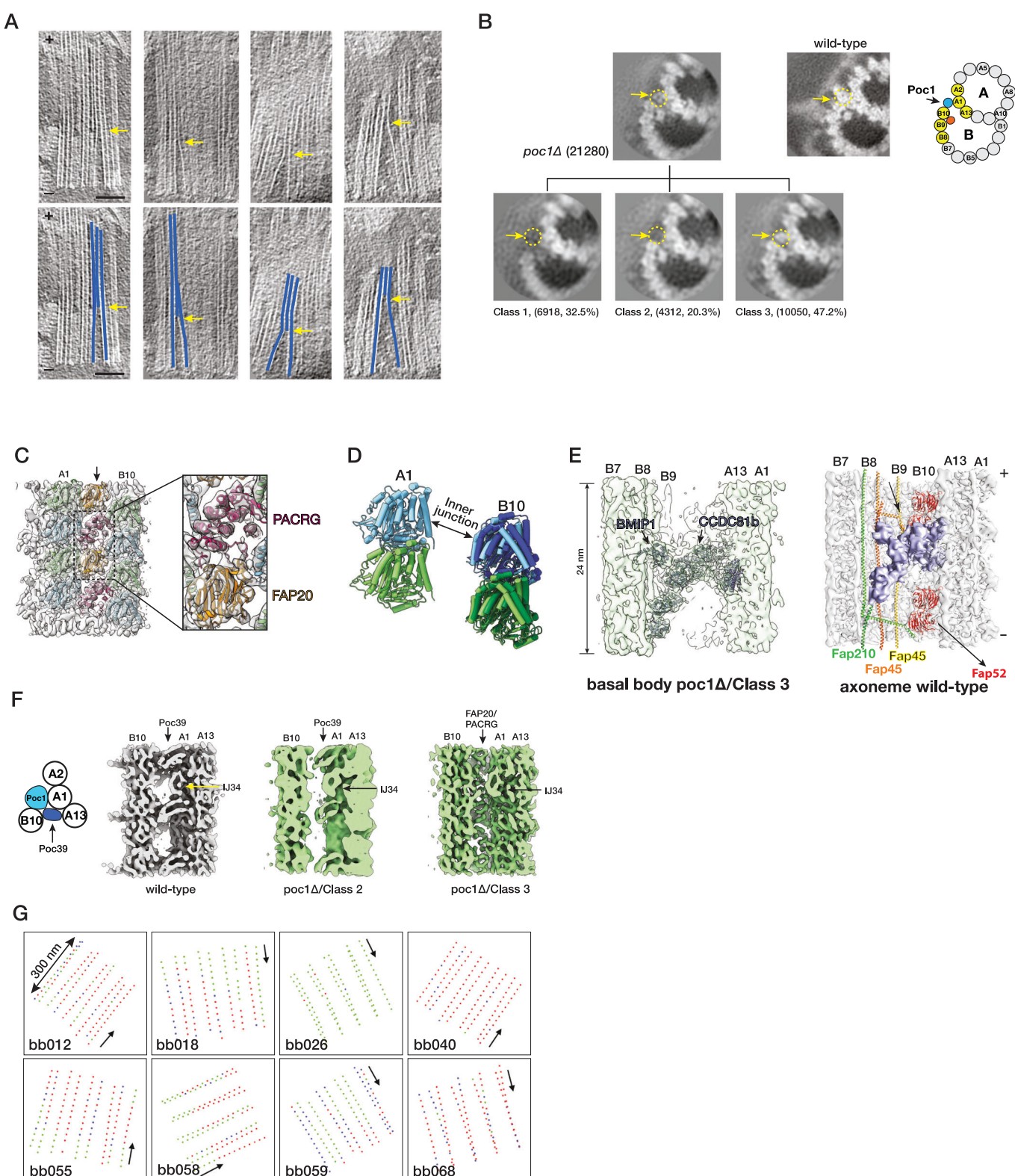

indicating a weakened interaction with its neighbors, resulting in a flexible and less well-defined inner junction without Poc1. Unexpectedly, we identified extra proteins in the third subset (Class 3, 47.2%) in the position of Poc1 (Fig. 5B). Further refinement of this subset improved the structure to 8.0 Å, resolving

the secondary structure of the protein folding (Fig. EV1F). This allowed us to identify these proteins as PACRG and FAP20 at the A–B inner junction (Fig. 5C; Movie EV8). Both PACRG and FAP20 are conserved axonemal inner junction components. However, they have not been found in the A–B inner junction in the central core

**Figure 5.  poc1Δ mutants destabilize the A–B inner junction and allow axonemal components to incorporate into the assembly.**

(A) Upper: representative tomogram slices from poc1Δ BB show the TMTs split at the A–B inner junction at various longitudinal locations. Bottom: the blue lines highlight the split TMTs in the upper panels. The yellow arrows indicate the locations of the split. The minus ends of TMT are at the bottom, and the plus ends are at the top. Scale bar: 100 nm. (B) 3D classification of subtomograms from poc1Δ BB, focusing on the A–B inner junction. The resulting three subsets show structure variations at the A–B inner junction. In Class 1, the B-tubule is incomplete and detached from the A-tubule at the inner junction. In Class 2, the B-tubule is complete and remains attached to the A-tubule, though the Poc1 position is empty. In Class 3, the B-tubule is complete and connected to the A-tubule, while other proteins occupy the Poc1 site. Yellow dashed circles and arrows outline the position of Poc1 in the wild-type. The numbers indicate the number of particles in each class and their percentage in the dataset. An image from the wild-type and a schematic diagram is shown for comparison. (C) Further refinement of Class 3 at 8.0 Å identifies the FAP20/PACRG filament at the mutant BB's inner junction gap. (D) Comparing the A–B inner junction in the central core region between wild-type and poc1Δ Class 3 shows an increased gap in the junction. Pf A1 is used as a reference for superposition. For Pf B10 (α/β tubulin), the wild-type is in light green and blue, and the poc1Δ is in dark green and blue. (E) Focused classification of subtomograms from poc1Δ BB (Class 3) identifies additional axonemal MIPs, CCDC81b, and BMIP1. Left, the mutant structure is shown in light green where the atomic models for CCDC81b/BMIP1 (PDB: 8V3I) are fit into the map. Right, the inner junction from the wild-type axoneme DMT is shown as a comparison. The corresponding MIPs CCDC81b/BMIP1, indicated by an arrow, are colored in lavender. (F) In poc1Δ BB, the LRR-motif MIP partially remains in the Class 2 average but is absent in Class 3, where the inner junction is occupied by FAP20/PACRG. For comparison, the wild-type BB structure is shown in gray. (G) Mapping the location of subtomogram from 3 subsets identified in the central core region of poc1Δ BBs. 8 representative BBs are shown. The green dots represent the Class 1 subset, an incomplete B-tubule detached from the A-tubule at the inner junction. The blue dots represent the Class 2 subset, complete B-tubule without Poc1. The red dots represent the Class 3 subset, the axoneme-like inner junction where FAP20/PACRG fills in the space left by Poc1. The arrows point in the direction from the proximal to the distal end of the BBs.

region of the wild-type BB (Ruehle et al, 2024). In addition to PACRG and FAP20, other MIPs, including IJ34, FAP106, FAP52, FAP45, and FAP210, could also be identified in this subset (Class 3) of the poc1Δ BB (Figs. 5F and EV5C). In contrast, the LRR-motif MIP was absent in this group (Fig. 5F). Further classification of this subset identified an additional MIP in 48-nm periodicity previously found unique to the inner junction in wild-type *Tetrahymena* axonemal DMT (Figs. 5E, EV1G, and EV5C) (Li et al, 2022). This MIP was proposed to be CCDC81b (Uniprot ID, I7M688) and BMIP1 (Uniprot ID, I7MB72) (Gao et al, 2024). Overlaying this mutant structure (Class 3) with the wild-type axoneme structure shows that these structures are nearly identical (Fig. EV5A,B, RMSD < 1 Å). In contrast, comparing this axonemal-like structure from poc1Δ BB to the wild-type central core region shows noticeable differences (Fig. 5D). In the poc1Δ mutant, the pf B10 is shifted (7.1 Å) and rotated (5.5°) relative to the A-tubule, resulting in an increased inner junction gap where the PACRG and FAP20 now fill in. Meanwhile, FAP52 moved closer to the A-tubule, similar to the movement from the central core region to the wild-type axoneme, as described in Fig. 4F. In summary, by incorporating axonemal components into the A–B inner junction, a subset (Class 3) of poc1Δ mutant BBs adopted a structure resembling the wild-type axonemal inner junction.

Finally, to find if there is any arrangement pattern for these three identified subsets, we mapped them to their corresponding locations in the core region for all 85 poc1Δ BB analyzed (Fig. 5G). A consistent pattern was not identified for any of these three subsets. Instead, the distribution shows sporadic dispersion throughout the longitudinal length of the central core region. For example, in one of the BBs (bb026), nearly all subtomograms have incomplete B-tubule at the inner junction. In contrast, in another BB (bb040), almost all inner junctions are found to be "axonemal-like" and were filled by PACRG/FAP20. However, in many BBs, the "axonemal-like" subtomograms aligned continuously, forming a single file in the same TMT, indicating a degree of cooperativity in PACRG/FAP20 assembly into the TMTs. It is conceivable that binding the first PACRG and FAP20 pair is the rate-limiting step, where the local geometry is not optimal for incorporation. However, once they are "wedged" into the inner junction, the local geometry, such as the gap between the pf B10 and A1 and the inter-protofilament curvature, becomes optimal, thereby facilitating recruiting additional PACRG/FAP20 pairs, manifested as cooperativity.

In summary, consistent with our recent study (Ruehle et al, 2024), we found that in the poc1Δ mutant, the assembly at the inner junction is disrupted, showing structural heterogeneity and composition variations. Without Poc1, the inner junction is structurally destabilized and accessible. This allows a set of axoneme components, including PACRG and FAP20, to be incorporated into the BB inner junction, leading to a change of the local MT lattice. These changes likely will propagate and have a long-range impact on the structural integrity of the TMT and BB.

## Discussion

In this work, we use cryoET and image analysis to study the structures of basal bodies and axonemes isolated from *Tetrahymena thermophila*. By focusing on the inner junction at three distinct locations, the proximal region, the central core region of the BB, and the axoneme region, we identified several MIPs present throughout the three locations in BBs and axonemes and proteins specific to a distinct region. These are summarized in Fig. 6A.

### MIPs shared by the BB and axoneme inner junction

Among the MIPs identified, FAP52 (or FAP52-like protein) and FAP106 are present in all three A–B inner junctions. This suggests that these MIPs are recruited to the TMTs at the start of BB biogenesis and are continuously assembled throughout the cilium. Both FAP106 (ENKUR in vertebrates) and FAP52 (WDR16 or CFAP52 in vertebrates) are highly conserved proteins found in all high-resolution axonemal DMT structures to date, ranging from the protozoans including Trypanosome (Shimogawa et al, 2023), *Chlamydomonas* (Ma et al, 2019), and *Tetrahymena* (Kubo et al, 2023; Li et al, 2022), to higher eukaryotes (Zhou et al, 2023; Walton et al, 2022; Leung et al, 2023). The proteins are associated with human ciliopathies (Ta-Shma et al, 2015; Sigg et al, 2017). Furthermore, FAP52 was implicated as a centrosome and BB component (Contreras and Hoyer-Fender, 2020; Hodges et al, 2010). This suggests essential roles for FAP52 and FAP106 in both centriole/BB biogenesis and motile cilium assembly.

Three additional MIPs previously identified in the axoneme have also been found at the A–B inner junction in the BB core region. These are IJ34, FAP45, and FAP210. While IJ34 is unique

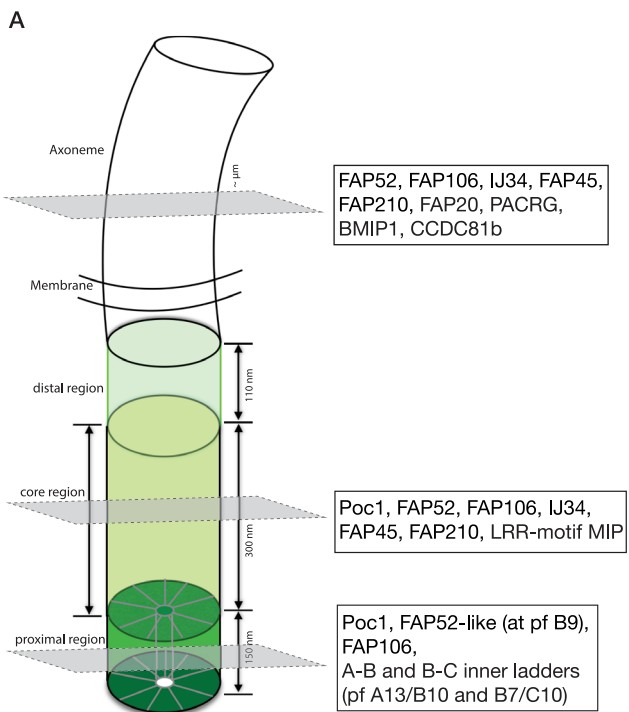

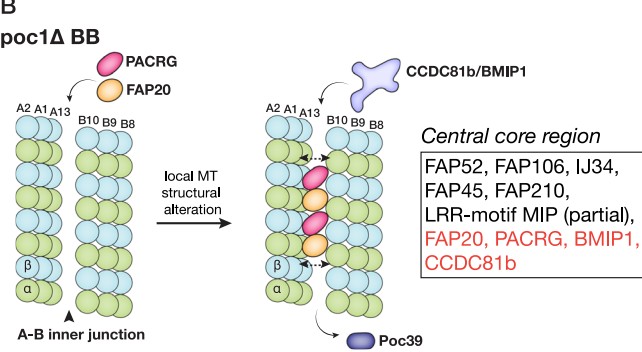

**Figure 6. Summary of inner junction MIPs localized in different regions of BB and axoneme and their misplacement in the poc1Δ mutant.**

(A) Schematic illustration summarizing the inner junction components in three regions of the cilium identified in this study. (B) A model illustrates the poc1Δ BB A–B inner junction partially morphing into an axoneme-like architecture. The double-ended arrows indicate the expansion of the inner junction gap once PACRG and FAP20 are incorporated. This causes disassociation of the LRR-motif MIP and facilitates the binding of axonemal components, such as CCDC81b and BMIP1. In the box, the axoneme-specific MIPs found in poc1Δ are indicated in red; note that many BB components are partially bound in the mutant as the inner junction structure is disrupted without Poc1.

continues to the axoneme. Furthermore, a recent study on *T. brucei* axoneme revealed FAP106's role as an "inner junction hub" critical for recruiting other MIPs, such as FAP45 and FAP210 (Shimogawa et al, 2023). Our identification of FAP106, FAP45 and FAP210 in the BB suggests that the hierarchical recruitment of MIPs found in the axoneme might also apply to the BB inner junction assembly.

The conservation of MIPs shared between the BB TMT and the axoneme DMT is likely not limited to the inner junction but extends to other parts of the A- and B-tubule (Ma et al, 2019), which we did not focus on in this work. Meanwhile, many previous phenotypical and functional analyses of MIPs have focused mainly on axonemes and flagella. However, their effects on the BBs have been grossly overlooked, mainly due to the relatively small size of the BB compared with the flagellum, structural polymorphisms, and the difficulty of biochemically isolating BBs. Given that many MIPs have structural roles in both BBs and flagellar axonemes, it is necessary to analyze both structures and their "connectome" to understand their impact on the cilium assembly and function. This is exemplified by a recent study on the axonemal protein CCDC39/40 (Brody et al, 2025), a heterodimer critical for recruiting other cilium components and dictating the axoneme's 96-nm periodicity. Deletion of CCDC39/40 not only results in the loss of over 90 ciliary proteins and disjointed ciliary structural organization, but the impacts extend to several cilia-independent pathways, such as protein homeostasis and cell differentiation.

## The ciliary inner junction MIPs are in a well-defined longitudinal location

The BB provides a template for forming a cilium. During its duplication, a new BB initially emerges as a probasal body (pBB) or procentriole. To make a mature BB capable of templating the cilium, the pBB first elongates to about 150 nm, forming the proximal region. This is followed by the sequential assembly of the central core region, the distal region, the transition zone, and, eventually, the axoneme (Allen, 1969). Each region is signified by its distinct ultrastructural features, for example, the cartwheel in the pBB, the inner scaffold in the central core region, the termination of C-tubule at the distal region, the Dynein arms and the central pair MT in the motile cilium axoneme. These morphological differences imply composition changes throughout the cilium assembly under a well-defined sequential control. Our study on the inner junctions, presented in molecular details in three distinct regions, agrees with this.

Interestingly, we found a FAP52-like protein binding to pf B9 in the proximal region while an unidentified density, the A–B inner ladder, associates to pf B10 and bridges pf B10 and A13. Longitudinally at ~130 nm, this FAP52-like MIP shifts location from pf B9 to FAP52$_{core}$ at pf B9/B10, coinciding with the termination of the A–B inner ladder, suggesting that these two events might be coordinated. Currently, we cannot confirm whether this FAP52-like MIP is FAP52 or its paralog gene products (UniProt IDs: I7MJ23, Q24C92, I7MLQ3) with similar tertiary structures (Kubo et al, 2023). Neither do we know how the shift of this FAP52-like MIP to FAP52 takes place at a longitudinal position of ~130 nm. It is plausible that other MIPs nearby, for example, the A–B ladder unique to the proximal region, could affect the binding site of this FAP52-like protein. Meanwhile, our AlphaFold 3

to *Tetrahymena* or perhaps other ciliates, FAP45 and FAP210 are conserved across phyla and have been found in *Tetrahymena*, *Chlamydomonas*, bovine, and in mouse and human sperm axonemal DMTs (Ma et al, 2019; Gui et al, 2021; Kubo et al, 2023; Zhou et al, 2023; Leung et al, 2023; Chen et al, 2023). Both are filamentous MIPs binding end-to-end and exhibiting 48-nm periodicity. They bind to TMT and DMT similarly in the BB and axoneme (Figs. 4A and EV4A, Movie EV4). This confirms that the internal 48 nm periodicity is established in the BB core region and

multimer calculation does not indicate FAP52 having a specific binding motif to the microtubule wall. Interestingly, the recent high-resolution structures of axonemal DMT implicated that FAP52 might preferentially bind to acetylated K40 of α-tubulin in the A/B inner junction (Ma et al, 2019; Khalifa et al, 2020). It is equally intriguing that ultrastructure expansion microscopy revealed the difference in acetylation state between human procentriole and mature centriole—while the mature centriole is fully acetylated, the procentriole, which becomes the proximal region of mature centriole later on, lacks the acetylation signal in its early assembly stage (Guichard et al, 2023). Hence, it is tempting to surmise that the difference in the MT acetylation stage might regulate the position of FAP52 or the FAP52-like MIP. Future study is needed to test these possibilities.

The structure change at ~130 nm is followed by the A–C linker's termination and the inner scaffold's emergence at ~170 nm, marking the transition from the proximal to the core region. In the core region, additional MIPs are recruited to the A–B inner junction, including FAP45, FAP210, IJ34, and the LRR-motif MIP, which are unique to this region. In the axoneme region, Poc1 is replaced by PACRG and FAP20, and the LRR-motif MIP is terminated while IJ34 remains. Coinciding with these composition changes, we observed MT structural alteration, likely to accommodate the changes, for example, variation in the local inter-protofilament lateral curvature and expansion of the lateral gap at the A–B inner junction in the axoneme.

The BB assembly initiates at its proximal end, where the TMT grows from the minus towards the plus direction (Allen, 1969). Our current study focused on the mature BBs that precluded assembly intermediates, such as pBBs. Thus, we are not able to provide any temporal resolution for the assembly process nor can we resolve relative timing when MIPs were incorporated into the BB. However, the spatial arrangement of the MIP observed in this study agrees well with the time-resolved gene expression profile in the ciliate *Stentor* (Sood et al, 2022). The study shows that the gene expressions involved in BB biogenesis and ciliary assembly are modular and can be described as a cascade wave of defined sets of genes. For example, POC1 is expressed before PACRG and FAP20, consistent with their sequential assembly in the BB and the axoneme. Similarly, a time-resolving RNA-seq study in *Tetrahymena* also indicates that expression of POC1 takes place before PACRG and FAP20 (Zhang et al, 2023). In summary, our observation of structures at three locations in the cilium is consistent with the notion that the assembly is a well-coordinated process under tight spatiotemporal control. However, a molecular understanding of the mechanism that governs this regulation is still incomplete.

### Axonemal proteins incorporated into BB inner junction in the absence of Poc1

Perhaps the most surprising finding in this study is identifying a set of axoneme-specific proteins, including PACRG and FAP20, in the poc1Δ mutant BB A–B inner junction. The inner junction in the wild-type BB core region and the axoneme share several structural features. Both have FAP52, FAP106, IJ34, FAP45, and FAP210 that bind to the MT wall in the same periodicity (Fig. 4A,B). However, the two inner junctions have marked differences. First, in the central core region, the A and B-tubules are crosslinked by Poc1, while in the axoneme, the location is filled by a PACRG/FAP20

filament. Second, there is a local geometry change. The lateral gap between pf A1 and pf B10 is wider in the axoneme compared to the BB core region (Fig. 4F). Third, the MTs exhibit different local inter-pf curvature (Fig. EV4F; Table 1). Fourth, the LRR-motif MIP is unique to the BB core region, while a set of MIPs, namely CCDC81b and BMIP1, are found only in the axoneme inner junction. The poc1Δ weakens the inner junction (Ruehle et al, 2024). The increased flexibility at the inner junction and the similarity of the local chemical environments to the axoneme will likely facilitate PACRG and FAP20 filling in the vacancy left by Poc1, leading to local structure changes. Once PACRG and FAP20 bind at the BB inner junction, it will promote recruiting additional PACRG and FAP20 and other axoneme components, such as CCDC81b and BMIP1. These structural changes result in a complete loss of the LRR-motif MIP in the inner junction. Thus, in the poc1Δ BB, the A–B inner junction partially morphs from a BB architecture to that of an axoneme (Fig. 6B).

Currently, we do not have a temporal resolution of these changes. Thus, we do not know the precise timing of PACRG/FAP20 incorporation in the poc1Δ BB inner junction. Nonetheless, given that the axoneme proteins PACRG/FAP20 are modularly expressed after the BB components (Sood et al, 2022; Zhang et al, 2023), it is likely that the incorporation takes place after the completion of BB assembly when PACRG/FAP20 congregate at the BB for cilium assembly (Yanagisawa et al, 2014). This implies that, in wild-type, the Poc1 assembled at the inner junction will prevent PACRG/FAP20 and other axonemal components from mistakenly being incorporated into the BB architecture. Interestingly, similar observations have been reported for other centriole or basal body mutants (O'Toole et al, 2003; Pudlowski et al, 2025; Sala et al, 2024). For example, a delta-tubulin deletion mutant in *Chlamydomonas* had its distal transition zone structure misplaced in the BB proximal region; in human cells, the loss of TEDC1 and TEDC2, which form a subcomplex with delta and epsilon-tubulin, led to extension of the proximal region proteins to the entire centriole. Perhaps this regulated sequential loading and replacement mechanism is a general method applied by the cell to maintain the spatial order of the BB components and to ensure the structural integrity of the organelle. In summary, the analysis of the inner junction structure in three regions, the proximal, the central core region of the BB and the axoneme, showcases the regulation of cilium biogenesis and provides structural insight into ciliary components, many of which can be linked to ciliopathies in humans.

## Methods

**Reagents and tools table**

| Reagent/resource | Reference or source | Identifier or catalog number |
| --- | --- | --- |
| **Experimental models** | | |
| Wild-type (*Tetrahymena t.*) | Tetrahymena Stock Center | B2086 (TSC_SD01625) |
| *poc1Δ* (*Tetrahymena t.*) | Pearson et al, 2009 | Dr. Pearson, Univ. Colorado, Denver |
| **Chemicals and reagents** | | |
| Phenyl methyl sulfonyl fluoride | Sigma | Cat P7626 |

| Reagent/resource | Reference or source | Identifier or catalog number |
|---|---|---|
| Proteinase Inhibitor Cocktail | Sigma | Cat P8340 |
| Igepal CA 630 | Sigma | Cat I8896 |
| TritonX-100 | Sigma | Cat T8787 |
| Quantifoil grid Cu 200 R2/2 | Ted Pella | PEL657-200-CU |
| Quantifoil grid Cu/Rh 200 R2/2 | Quantifoil, Inc | N/A |
| BSA-coated 10 nm gold | BBI | Cat EM.BSA10/2 |
| **Software** | | |
| MotionCor2 | SBGrid (https://sbgrid.org/) | N/A |
| IMOD | SBGrid (https://sbgrid.org/) | N/A |
| TomoAlign/TomoRec | SBGrid (https://sbgrid.org/) | N/A |
| TomoCTF | SBGrid (https://sbgrid.org/) | N/A |
| MLTOMO/Xmipp | SBGrid (https://sbgrid.org/) | N/A |
| Spider | SBGrid (https://sbgrid.org/) | N/A |
| RANSAC | https://tiny.cc/ransac | N/A |
| Relion 3.1 | SBGrid (https://sbgrid.org/) | N/A |
| Relion 4.0 | SBGrid (https://sbgrid.org/) | N/A |
| Alphafold2 | SBGrid (https://sbgrid.org/) | N/A |
| Alphafold3 | https://alphafoldserver.com/welcome | N/A |
| ChimeraX | https://www.rbvi.ucsf.edu/chimerax/ | N/A |
| **Other** | | |
| Titan Krios microscope | Thermo Fisher Scientific | N/A |
| K2 Detector | Gatan, Inc | N/A |
| K3 Detector | Gatan, Inc | N/A |
| Bio-Quantum GIF | Gatan, Inc | N/A |

## Sample and EM grid preparations

The axoneme and the BB isolates from the *Tetrahymena* wild-type and poc1Δ strain have been described in detail (Li et al, 2022; Ruehle et al, 2024). The wild-type (B2086) *Tetrahymena* strain was obtained from the Tetrahymena Stock Center at Cornell University. The poc1Δ strain was generated as in (Pearson et al, 2009).

The BB and axoneme EM grids were made at the University of Colorado Anschutz Medical Campus (Aurora, CO) or the University of California in Davis (Davis, CA), respectively. In total, 200 mesh Quantifoil Cu or Cu/Rh 2/2 (Quantifoil, Inc) were used for all samples. A 4 μl sample mixed with 10 nm colloid gold

beads coated with BSA was applied to the grid and plunge frozen in liquid ethane after various amounts of wait time (10–50 s.) using a Vitrobot (Thermo Fisher, Inc). The relative humidity in the Vitrobot chamber was 95%, the temperature was 22 °C and the blot time was 0.5–1.0 s. The frozen grids were stored in liquid nitrogen and were transported to UCSF for data collection.

## Electron cryo-tomography data collection

Single-axis tilt series were collected for all samples on two field emission guns, 300 kV Titan Krios electron microscopes (Thermo Fisher, Inc) at UCSF. Each scope had a Bio-Quantum GIF energy filter and a post-GIF Gatan K2 or K3 Summit Direct Electron Detectors (Gatan, Inc.). The GIF slit width was set at 20 eV. SerialEM was used for tomography tilt series data collection (Mastronarde, 2005). The data were collected in the super-resolution and dose-fractionation mode. The nominal magnification was set at 33,000. The physical pixel size on recorded images was either 2.70 Å (for wild-type BB) or 2.65 Å (for poc1Δ mutant and wild-type axoneme). A dose rate was set at 20 electrons/pixel/second during exposure. A bi-directional scheme was used for collecting tilt series, starting from zero degrees, first tilted towards -60°, followed by a second half from +2° to +60°, in increments of 2° per tilt. The accumulated dose for each tilt series was limited to 80 electrons/Å$^2$ on the sample.

## EM data processing and image analysis

For the tilt series alignment and tomogram reconstruction, the dose-fractionated movie at each tilt in the tilt series was corrected of motion and summed using MotionCor2 (Zheng et al, 2017). The tilt series were aligned using the gold beads as fiducials using IMOD and TomoAlign (Fernandez et al, 2018). The contrast transfer function for each tilt series was determined and corrected by TomoCTF (Fernández et al, 2006). The tomograms were reconstructed by TomoRec, which considered the beam-induced sample deformation during data collection (Fernandez et al, 2019). A total of 156 wild-type BB tomograms, 85 poc1Δ BB tomograms, and 51 wild-type axoneme tomograms were used for reconstruction.

For subtomogram alignment and averaging, the BBs or the axoneme were first identified in the 6xbinned tomograms. The center of TMT or DMT and their approximate orientation relative to the tilt axis were manually annotated in a Spider metadata file (Frank et al, 1996). The initial alignment and average were carried out in a 2× binned format (pixel 5.4 Å for the wild-type or 5.30 Å for poc1Δ mutant and axoneme). The longitudinal segment length of TMT or DMT in a subtomogram was limited to 24 nm and 50% overlapping with neighboring segments. In all cases, subtomogram alignments were carried out without using any external reference by a program MLTOMO implemented in the Xmipp software package (Scheres et al, 2009).

Since TMTs and DMTs are continuous filaments, a homemade program, RANSAC (available at https://tiny.cc/ransac), was used to detect any alignment outliers and impose the continuity constraint on the neighboring segments after obtaining the initial alignment parameters. This corrected the misaligned subtomograms if only a few subtomograms within the same filament were misaligned. Otherwise, the entire filament was discarded without further processing. MLTOMO and Relion v3.1 or v4.0 were extensively used for the focused classification of the subtomograms (Bharat and Scheres, 2016; Zivanov et al, 2022). Customized soft-edge binary masks were used during classification to limit the analysis to the

**Table 2. Summary of structures.**

| Structure | Number of particles | Resolution (Å) | Description | EMDB ID # |
|---|---|---|---|---|
| 1 | 12290 | 9.8 | A–B inner junction in the proximal region, 16 nm, wild-type BB | EMD-46437 |
| 2 | 7665 | 9.8 | B–C inner junction in the core region, 16 nm, wild-type BB | EMD-46438 |
| 3 | 4664 | 9.3 | A–B inner junction in the core region, 48 nm, wild-type BB | EMD-46439 |
| 4 | 34006 | 8.3 | A–B inner junction in the core region, 16 nm, wild-type BB | EMD-46440 |
| 5 | 15473 | 5.9 | A–B inner junction in the axoneme, 16 nm, wild-type | EMD-46441 |
| 6 | 10050 | 8.0 | A–B inner junction in the core region, 16 nm, poc1Δ BB | EMD-46442 |
| 7 | 2315 | 10.1 | A–B inner junction in the core region, 48 nm, poc1Δ BB | EMD-46443 |

structure of interest. This was critical for determining the correct periodicity of the MIPs and identifying structural defects or heterogeneity in the structure. We found the MIPs in the BB proximal region have up to 16-nm periodicity, MIPs in the BB central core region and the axoneme have up to 48-nm periodicity. Once the correct periodicity was found, these out-of-register subtomograms were re-centered and re-extracted. This was followed by combining all subtomograms for the next round of refinement. Previously, we found the MIPs at the inner junction in the core region of BB having 16-nm periodicity. It was based on analysis limited to a small cross-section region, including pfs B9, B10, A1, and A13. In this study, we expanded to a larger area, including more pfs. This led us to the identification of filamentous MIPs (fMIPs) FAP45 and FAP210 in the BB central core region. Both fMIPs have a 48-nm periodicity, the same as in the axonemes (Ma et al, 2019; Kubo et al, 2023; Zhou et al, 2023; Gui et al, 2021; Leung et al, 2023). The final refinements, focusing on the inner junctions, were implemented using a workflow in Relion v4.0 (Zivanov et al, 2022). The overall resolutions (Fig. EV1; Table 2) are based on the Fourier Shell Correlation (FSC) cutoff at 0.143 (Rosenthal and Henderson, 2003; Scheres and Chen, 2012). Care was taken to ensure the two-half datasets were independent without any overlap. A generous soft-edge mask was used when calculating the FSC. The number of particles used for the final averaged structure can be found in Table 2.

Due to extensive heterogeneity and a limited number of particles for averaging, the B–C inner junction and the distal region of the basal body are excluded from this study.

### Model building and analysis

The pseudo-atomic models for the inner junctions in different regions of the BB and the axoneme were built based on the subtomograms-averaged density maps described in this work. UCSF ChimeraX (Pettersen et al, 2021) was used for model building. The atomic models for individual MIP were based on previously published data (PDB: 8G2Z, 8V3I) (Kubo et al, 2023; Gao et al, 2024) or models predicted by AlphaFold2 (Jumper et al, 2021), which was installed and run locally on the UCSF Wynton HPC, or AlphaFold 3 (Abramson et al, 2024) via the AlphaFold server (https://alphafoldserver.com/welcome). The models for individual MIP or α/β tubulins were fit into the subtomogram averaging density maps in subnanometer resolution, using the "FitMap" command in UCSF ChimeraX. Two constraints were applied when fitting the MIPs whose atomic models were available from the previous axoneme structure, the location and the tertiary fold of a specific MIP. For the AlphaFold 3 predicted structure,

a ChimeraX built-in function "AlphaFold error plot" was used to color the predicted model based on its Predicted Aligned Error (PAE) score provided by the AlphaFold server. UCSF ChimeraX was also used for visualization and for recording images and movies. The pseudo-atomic models can be obtained from the authors upon request.

To analyze the microtubule inter-protofilament angle and curvature, the UCSF ChimeraX command "matchmaker" was first used to bring two reference tubulin models to the two targets in the neighboring protofilaments. This was followed by the "measure rotation" command to find the rotation angle between the two reference models. The output is the inter-protofilament angle, a local MT lateral curvature measurement.

To measure the difference of inner junction width between pfs A1 and B10 at the regions of BB or axoneme, once the MT models were built, ChimeraX command "matchmaker" was used to bring the two models to the same reference; in this case, the pf A1 served as the reference. This is followed by a second command, "rmsd #Model1@ca to #Model2@ca". This provided the root-mean-square deviation (RMSD) between the two molecules in pf B10, measuring the width change between the inner junctions.

### Tetrahymena gene expression profile

The gene expression profiles were obtained via the Tetrahymena Genome Database website (https://tet.ciliate.org/index.php). The following *Tetrahymena* genes are used for the query: POC1 (TTHERM_01308010), FAP20 (TTHERM_00418580), PACRGA (TTHERM_00446290), PACRGB (TTHERM_00499570), PACRGC (TTHERM_00499310).

## Data availability

The EM structures have been deposited in the EMDB with the following accession numbers: EMD-46437, EMD-46438, EMD-46439, EMD-46440, EMD-46441, EMD-46442, EMD-46443.

The source data of this paper are collected in the following database record: biostudies:S-SCDT-10_1038-S44318-025-00392-6.

## Peer review information

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

## Acknowledgements

The authors thank Brian Wimberly, Dave Farrell, Peter Van Blerkom, Eduardo Romero Camacho (University of Colorado School of Medicine, Denver), and Fei Guo (University of California, Davis) for help with screening and preparing EM grids; Annie Rhee (University of California, Davis) for technical assistance. The authors thank David Bulkley and Eric Tse (UCSF) for assistance on tomography data collection, the Wynton HPC team (UCSF) for supporting the computational infrastructure, Tom Goddard (UCSF) for help with AlphaFold2 and UCSF ChimeraX software. The UC Davis BioEM Facility is supported by user fees, the Department of Molecular and Cellular Biology, the College of Biosciences, the Office of Research and the Provost's Office. This work is supported in part by NIH grants GM127571 (MEW), GM118099 (DAA), GM140813 (CGP), and by the Spanish AEI/FEDER (PID2022-139071NB-I00) (J-JF).

## Author contributions

**Sam Li**: Conceptualization; Data curation; Software; Formal analysis; Validation; Investigation; Visualization; Methodology; Writing—original draft; Writing—review and editing. **Jose-Jesus Fernandez**: Software; Formal analysis; Validation; Methodology; Writing—review and editing. **Marisa D Ruehle**: Data curation; Investigation; Writing—review and editing. **Rachel A Howard-Till**: Data curation; Writing—review and editing. **Amy Fabritius**: Data curation; Writing—review and editing. **Chad G Pearson**: Conceptualization; Resources; Supervision; Funding acquisition; Project administration; Writing—review and editing. **David A Agard**: Resources; Funding acquisition; Methodology. **Mark E Winey**: Conceptualization; Resources; Supervision; Funding acquisition; Investigation; Project administration; Writing—review and editing.

Source data underlying figure panels in this paper may have individual authorship assigned. Where available, figure panel/source data authorship is listed in the following database record: biostudies:S-SCDT-10_1038-S44318-025-00392-6.

## Disclosure and competing interests statement

The authors declare no competing interests.

# Expanded View Figures

**Figure EV1.  Related to Figs. 1, 2, 4, 5, S5. Assessing resolution of subtomogram averages by Fourier shell correlation (FSC).** ▶

The structures and their Fourier shell correlations as a function of resolution (1/Å) are reported in Table 2. (**A**) A 16-nm repeat of the A–B inner junction from the proximal region of BB (wild-type). (**B**) A 16-nm repeat of the B–C inner junction from the proximal region of BB (wild-type). (**C**) A 48-nm repeat of the A–B inner junction from the central core region of BB (wild-type). (**D**) A 16-nm repeat of the A–B inner junction from the central core region of BB (wild-type). (**E**) A 16-nm repeat of the A–B inner junction from the axoneme (wild-type). (**F**) A 16-nm repeat of the A–B inner junction from a subset (Class 3) of the central core region of poc1Δ BB. (**G**) A 48-nm repeat of the A–B inner junction from a subset (Class 3) of the central core region of poc1Δ BB.

**A**. 16-nm repeat of the A–B inner junction from the proximal region of basal body (wild-type)

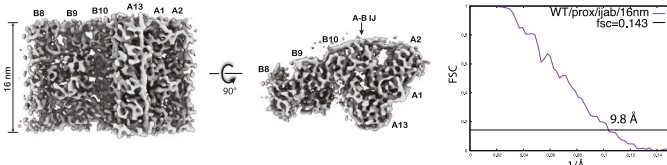

**B**. 16-nm repeat of the B–C inner junction from the proximal region of basal body (wild-type)

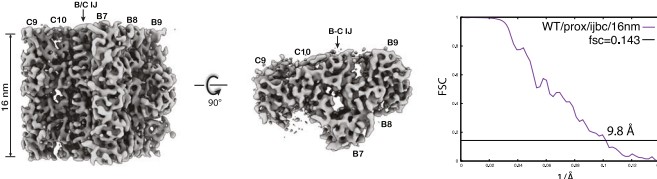

**C**. 48-nm repeat of the A–B inner junction from the central core region of basal body (wild-type)

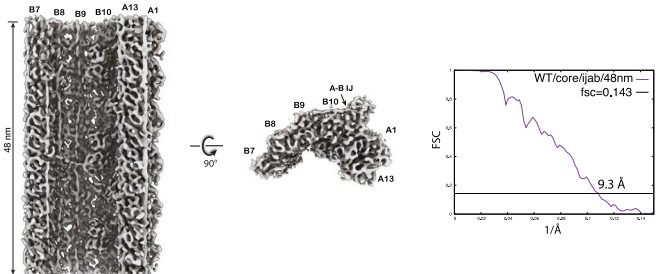

**D**. 16-nm repeat of the A–B inner junction from the central core region of basal body (wild-type)

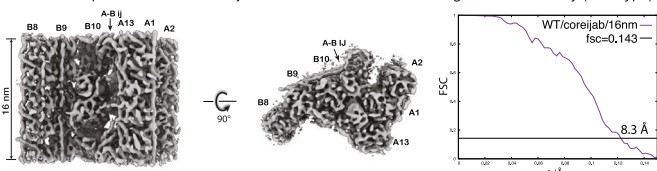

**E**. 16-nm repeat of the A–B inner junction from the axoneme (wild-type)

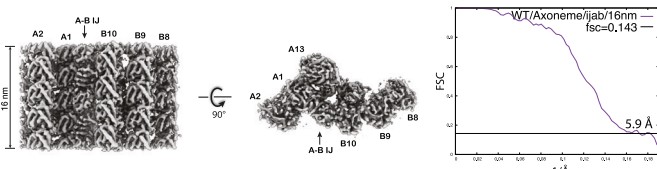

**F**. 16-nm repeat of the A–B inner junction from the central core region of a subset (Class 3) of poc1Δ BB

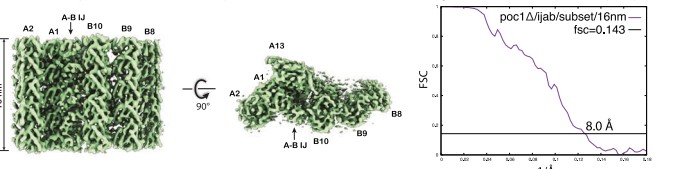

**G**. 48-nm repeat of the A–B inner junction from the central core region of a subset (Class 3) of poc1Δ BB

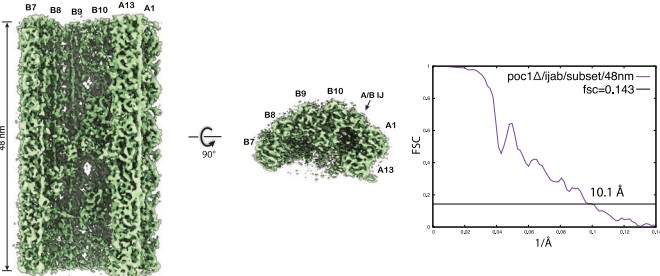

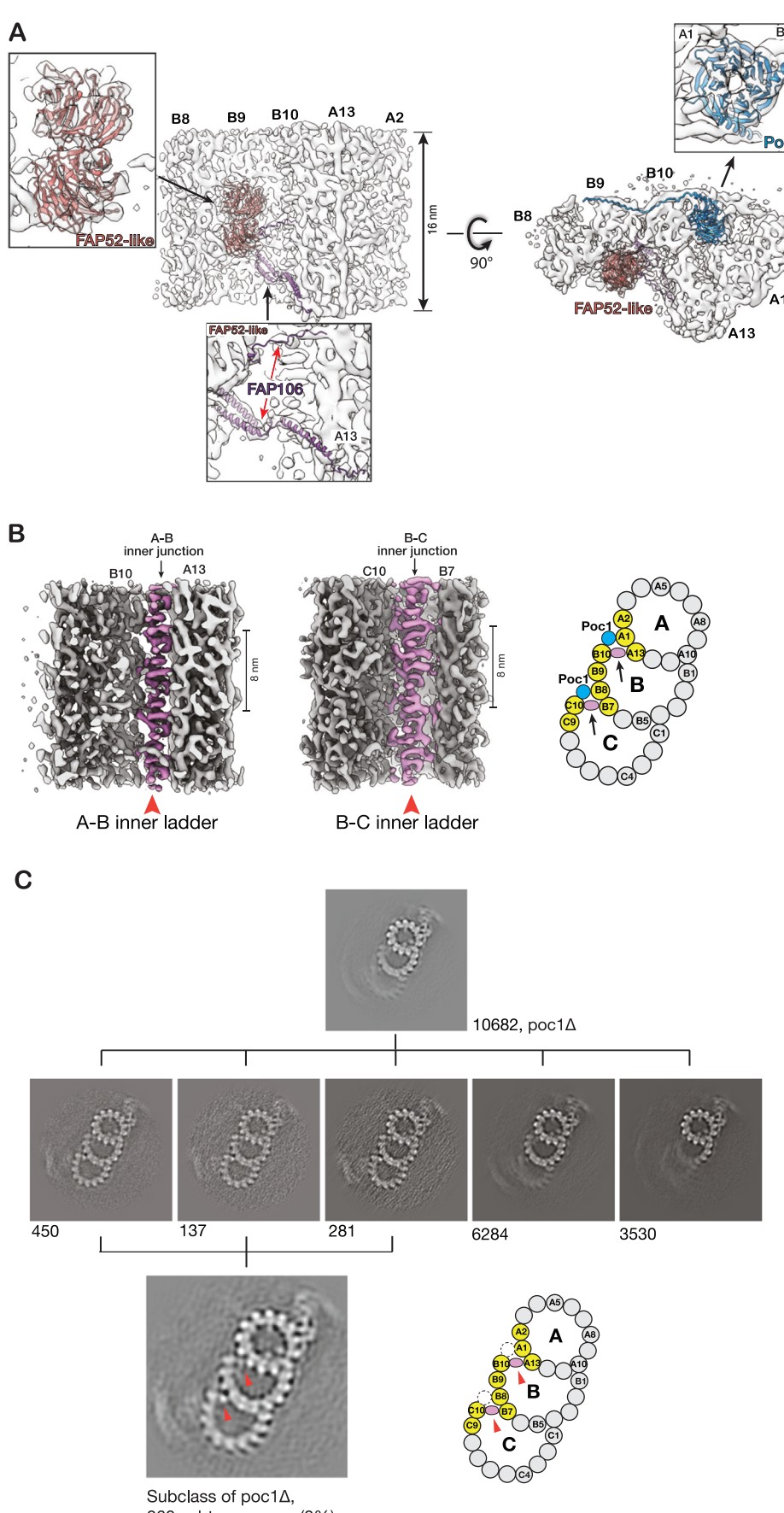

◀

**Figure EV2. Related to Fig. 2. The inner junctions in the proximal region.**

(A) Fitting the atomic models of FAP52, FAP106 and Poc1 into the 16-nm repeat averaged density map from the proximal region of BB. The FAP52 and FAP106 models are based on the axoneme structure (PDB: 8G2Z). The Poc1 model is from the AlphaFold2 database. (B) The maps of the A–B and B–C inner junctions in the proximal region of BB. The two unidentified proteins, namely the A–B inner ladder and B–C inner ladder crosslinking pfs A13-B10 or B7-C10, respectively, are highlighted in light pink and indicated by red arrowheads. On the right, a schematic diagram shows the location of the above two maps in the TMT. Black arrows indicate the viewing directions. (C) 3D Classification of the subtomograms from the proximal region of poc1Δ TMT identified a small fraction of the dataset (8%) having complete TMT, where the two unidentified proteins, the A–B inner ladder and B–C inner ladder indicated by red arrowheads, remain in the inner junctions. The number of subtomograms in each class is shown. The dashed circles in the cartoon indicate the location of Poc1 in the wild-type.

A

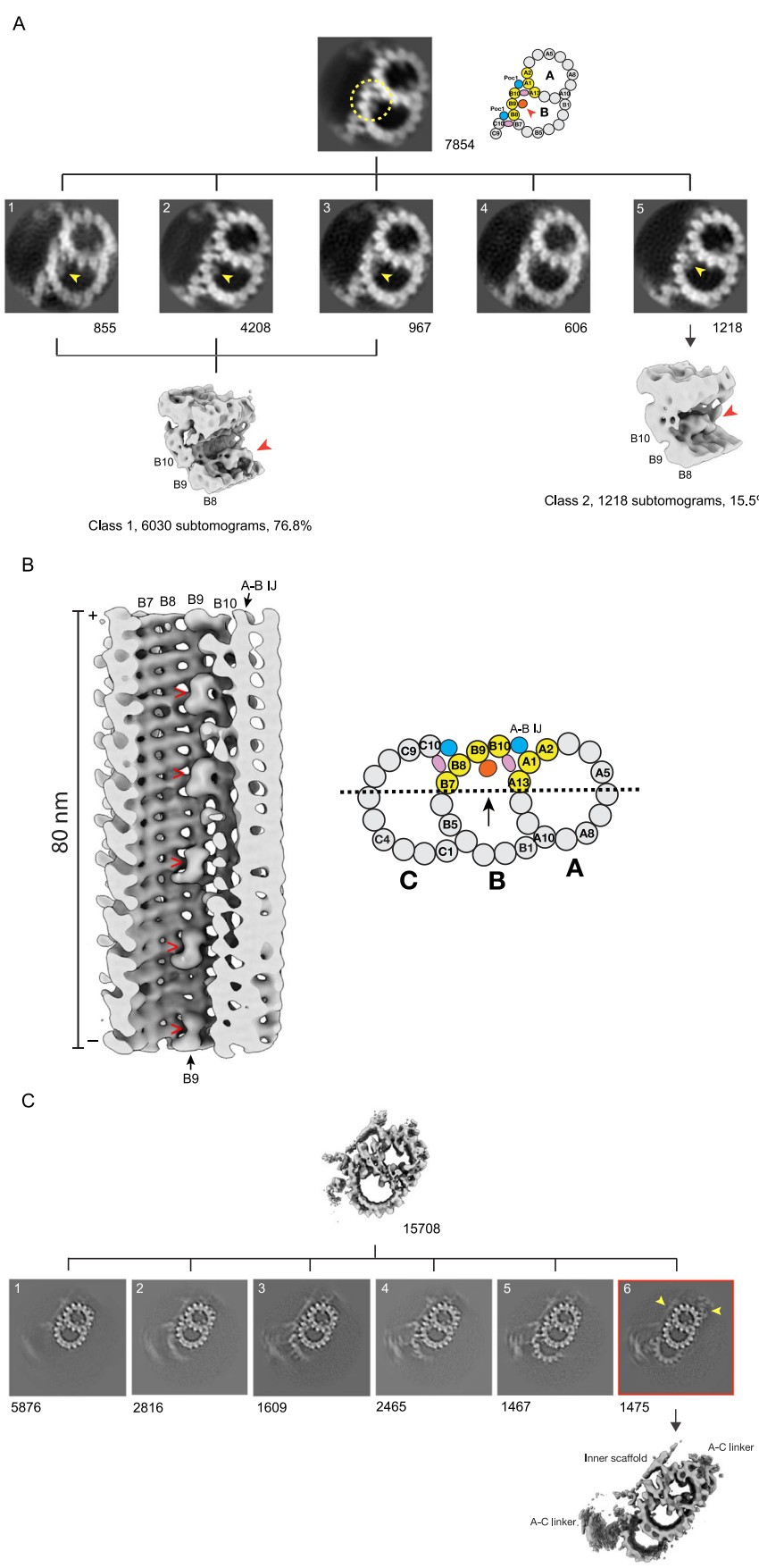

Class 1, 6030 subtomograms, 76.8%

Class 2, 1218 subtomograms, 15.5%

B

80 nm

B7 B8 B9 B10

A-B IJ

B9

C

15708

Inner scaffold

A-C linker

A-C linker

1475 subtomograms, 9.4%

◄ **Figure EV3.   Related to Fig. 3.**

(**A**) Focused 3D classification on the subtomograms from the proximal region of the BB. A yellow dashed circle indicates the focused area centered on the inner junction. Yellow or red arrowheads indicate the locations of FAP52 in the class averages. (**B**) A Longitudinal cross-section of the A–B inner junction showing the FAP52, indicated by red arrowheads, shifts binding from pf B9 to pf 9/10. A schematic illustration of the TMT is on the right. A dashed line and an arrow indicate the cross-section and the viewing direction of the structure on the left. A red dot represents FAP52. (**C**) Focused 3D classification on the subtomograms from the central core region of the BB. The Class 6 is highlighted with a red frame. In this class, the A–C linker and the inner scaffold are indicated by yellow arrowheads. The number of subtomograms in each group is provided.

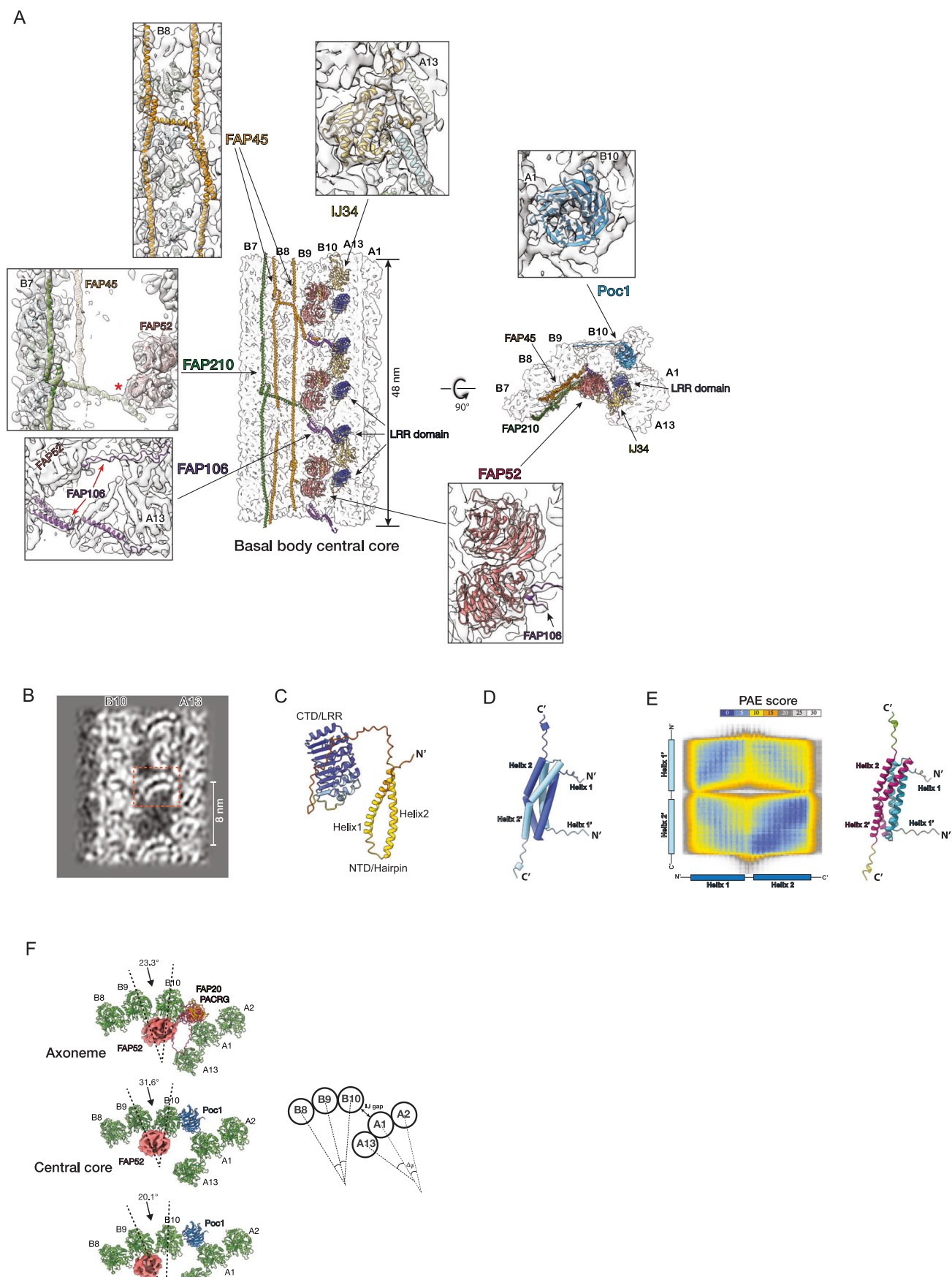

**Figure EV4. Related to Fig. 4.**

(A) Fitting the atomic models of MIPs found in the central core region of BB into the averaged density map. Each inset shows the fitting of a MIP model into its local density and its surroundings. In the "FAP210" inset panel, a red asterisk indicates a potential interaction between FAP210 and FAP52 that has been observed previously in the axoneme structure. In the "FAP52" inset panel, a black arrow indicates a potential interaction between FAP52 and FAP106. The α/β tubulins are colored in pale green and blue in the background. The 16-nm repeat map from the central core region is used for fitting of FAP52. FAP106, IJ34, and Poc1 models, as these MIPs have 16-nm or 8-nm (Poc1) periodicity. (B) A cross-section slice of the density map shows an LRR motif highlighted in a red dashed line square. (C) An AlphaFold2 predicted protein structure (UniProt Q22N53) was identified previously in the BB proteome. The protein is composed of a N-terminal α-helix hairpin (NTD) and a C-terminal LRR motif (CTD). The structure is colored based on the prediction confidence score (pLDDT: the predicted local distance difference test). The high confidence is in dark blue, while the low confidence is in yellow or orange. (D) An AlphaFold 3 predicted 4-helix bundle formed by dimerizing two NTD/hairpins. Two copies of the NTD hairpin from the LRR-motif MIP (UniProt Q22N53) form an anti-parallel dimer. One monomer is in dark blue (Helix 1 and Helix 2) and the other monomer is in light blue (Helix 1' and Helix 2'). The dimer forms a right-handed 4-helix bundle. (E) The predicted aligned error (PAE) plot provides inter-domain packing confidence scores. The dark and light blue imply the prediction with high confidence, while the gray and white indicate low confidence in the interaction. On the right, a ChimeraX-adapted color scheme is used where the 4-helix bundle is colored based on the PAE potential interaction score. The two N-terminal helices (Helix 1, Helix 1') are in cyan, and the two C-terminal helices (Helix 2 and Helix 2') are in magenta, indicating that the interaction between Helix 1 and Helix 1' (cyan), Helix 2 and Helix 2' (magenta) are with high confidence. (F) Inter-protofilament angle measurement. Left: the angles between pfs B9 and B10 are measured at the three regions, showing the variation of local curvature. The FAP52 and Poc1 or FAP20/PACRG are shown as reference points. Right: a schematic diagram depicts the inter-protofilament angles at the A–B inner junction. More complete measurements are in Table 1.

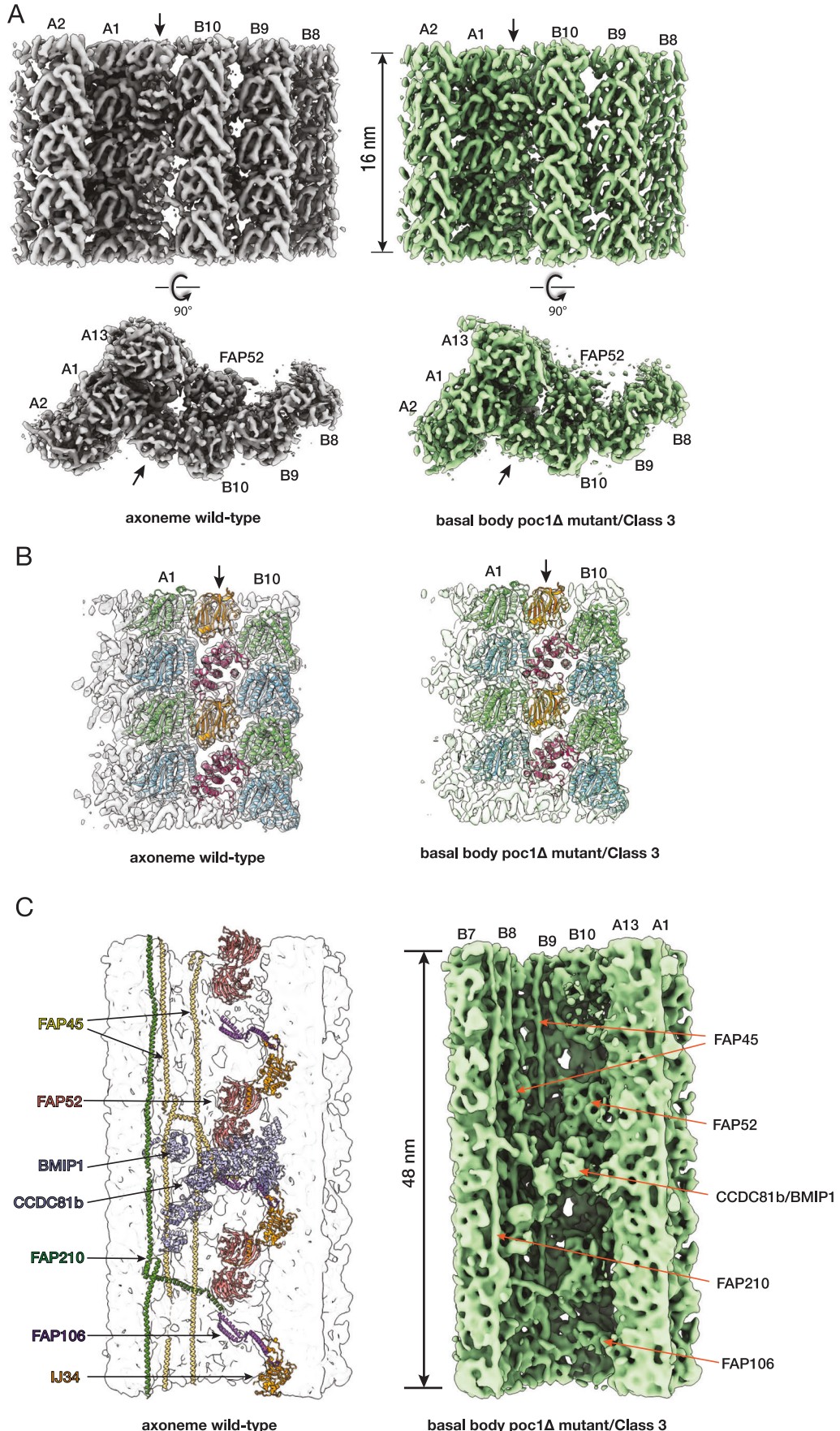

◀

**Figure EV5. Related to Fig. 5. Comparing the inner junctions between the wild-type axoneme and the Class 3 from the poc1Δ BBs.**

(A) Comparing the two structures in two orthogonal views, the wild-type axoneme is in gray, and the poc1Δ BB is in green. (B) Fitting the molecular models into the density maps in (A). The α/β tubulins are in light green and blue, FAP20 is in orange, and PACRG is in dark magenta. The arrows indicate the A–B inner junctions. (C) Comparing the 48-nm repeat from the wild-type axoneme (left) and from a subset of poc1Δ BB (Class 3, on the right in green). The 48-nm repeat average from a subset of poc1Δ BB (Class 3) shows nearly identical structure to the wild-type axoneme inner junction. This is the same subset/class as shown in Fig. 5E, but the longitudinal length is extended to 48 nm here. For clarity, the poc1Δ BB mutant map is low-pass filtered to 12 Å.

