## [Peer Review File · The EMBO Journal]

The Structure of Basal Body Inner Junctions from Tetrahymena Revealed by Electron Cryo-Tomography

Sam Li, Jose-Jesus Fernandez, Marisa Ruehle, Rachel Howard-Till, Amy Fabritius, Chad Pearson, David Agard, and Mark Winey

Corresponding author(s): Sam Li (samli@msg.ucsf.edu) , Mark Winey (mwiney@ucdavis.edu), David Agard (david.agard@czii.org)

Review Timeline:

Submission Date:	16th Sep 24
Editorial Decision:	15th Oct 24
Revision Received:	8th Jan 25
Accepted:	10th Feb 25

Editor: Hartmut Vodermaier

Transaction Report:

Dr. Mark Winey
University of California, Davis
Department of Molecular and Cellular Biology
Davis, CA 95616

15th Oct 2024

Re: EMBOJ-2024-119050-T
The Structure of Cilium Inner Junctions Revealed by Electron Cryo-tomography

Dear Dr. Winey,

Thank you for submitting your cryo-ET study on basal body inner junctions to The EMBO Journal. It has now been assessed by three expert referees, whose comments are copied below for your information. As you will see, all referees appreciate the new analyses and find the structural insights potentially interesting. However, they also raise several experimental and presentational points that would require improvement prior to publication in The EMBO Journal.

Should you be able to adequately address the key concerns of the reviewers, we would be interested in pursuing a revised version further for EMBO Journal publication. Since it is our policy to allow only a single round of major revision, it will however be important to carefully respond to all points at the time of resubmission. Also, please do not hesitate to contact me already during the early stages of the revision work, in case you would like discuss how to best tackle particular issues. Finally, should you require more time than our default three-months revision period, we would be happy to offer an extension, during which our 'scooping protection' (meaning that competing work appearing elsewhere in the meantime will not affect our considerations of your study) would of course remain valid.

Further information on preparing and uploading a revised manuscript can be found below and in our Guide to Authors. Thank you again for the opportunity to consider this work for The EMBO Journal, and I look forward to your revision.

Yours sincerely,

Hartmut Vodermaier

9) To facilitate reproducibility and cross-laboratory adoption of methodologies, please structure the Materials & Methods section as outlined in our guide to authors, including a completed Reagents and Tools Table that can be downloaded from our author guidelines as well (<https://www.embopress.org/page/journal/14602075/authorguide#structuredmethods>).

10) Digital image enhancement is acceptable practice, as long as it accurately represents the original data and conforms to community standards. If a figure has been subjected to significant electronic manipulation, this must be clearly noted in the figure legend and/or the 'Materials and Methods' section. The editors reserve the right to request original versions of figures and the original images that were used to assemble the figure. Finally, we generally encourage uploading of numerical as well as gel/blot image source data; for details see: embopress.org/page/journal/14602075/authorguide#sourcedata

At EMBO Press, we ask authors to provide source data for the main manuscript figures. Our source data coordinator will contact you to discuss which figure panels we would need source data for and will also provide you with helpful tips on how to upload and organize the files.

In the interest of ensuring the conceptual advance provided by the work, we recommend submitting a revision within 3 months (13th Jan 2025). Please discuss the revision progress ahead of this time with the editor if you require more time to complete the revisions. Use the link below to submit your revision:

Link Not Available

Referee #1:

Li and colleagues reconstructed 3D structure of the basal body (BB) from *Tetrahymena* at ~9Å resolution. With this pseudo-atomic resolution and referring to known atomic structure of *Chlamydomonas* axoneme (by the Brown group), they identified molecules at the inner junction of triplet microtubules (BB TMT) and built models of three flagellar regions - proximal and core parts of BB (170nm and 300nm length, respectively), as well as the axoneme. They assigned one internal density of TMT to FAP52, a known microtubule internal protein (MIP) of the axoneme. This FAP52 is located differently between the proximal and central core parts of BB. While the location at the central core is similar to that in the axoneme, FAP52 is shifted towards an adjacent microtubule protofilament. Another highlighted protein is Poc1. In their reconstructions, Poc1 is located at the inner junction between A- and B-tubules and between B- and C-tubules. Poc1 knockout induces loss of other MIPs, according to their subtomogram classification, and causes distortion of BB.

While this reviewer appreciates their novel findings, which will encourage the cilia/centriole community for further functional and structural studies on these proteins, a few clarifications should be done. After the authors' addressing the following points, this reviewer will support publication in the EMBO Journal.

Major points:

Despite the well fitted FAP52 atomic model to their cryo-ET density map, at this resolution, possibility that this density is from another protein, which has similar structure as FAP52, cannot be excluded. Even if experimental proof, such as genetic tagging, is not realistic, the authors could still make their assignment more convincing by AlphaFold-multimer to demonstrate the two types (proximal BB and axoneme) of tubulin-FAP52 interaction, or by proving the absence of similar folding prediction from the

Tetrahymena genome or BB proteomics.

They classified subtomograms and showed two ways of FAP52 location/orientation gradually altering from the proximal to the central parts of BB and also two classes. A similar plot is shown for presence and absence of inner scaffold. How are distributions of these multi-structures in BB? It should be possible to show presentation, similarly to Fig.S5G to examine how this transition occurs.

In their previous work (Ruehle et al. 2024), three regions (proximal, central and distal) parts of BB were compared. In this manuscript, the 100nm length distal region is not mentioned and they compare proximal and central BB regions and the axoneme. It will be fair to show structure of the third BB region. Or is the distal region not involved in BB by their preparation?

Minor points:

p.19, line 8: Please refer a literature which proved that the proximal region precedes the core region in the BB generation.

p.20, line 3: In addition to structural alteration, does BB undergo angle change? Twist is observed during transition from BB to the axoneme. Does this twist occur within the BB?

Fig.2B: Difference between upper and lower panels should be explained.

p.34, line6: RANSAC must be explained. Is there a literature?

Table 1: Are the number of particles mentioned here the number finally used for averaging? If so, how many particles did the authors start classification/averaging? How were the resolution calculated - which mask did they use to calculate FSC "at the A-B inner junction", for example?

Referee #2:

The article by Sam Li and colleagues presents a structural analysis of the inner junction of the triplet microtubule within the basal body and cilium. This inner junction refers to the closure of the B or C microtubule with the A or B microtubule, respectively. Utilizing sub-tomogram averaging, which achieves sub-nanometric resolution across three distinct regions - the proximal region of the basal body, the central core, and the axoneme - the authors conducted a comparative analysis that reveals structural variations along the triplet microtubule of the basal body across these regions. The quality of the work is high, offering valuable insights into the field of basal body and cilium research. Below are some comments for the authors to consider before publication:

- The title does not accurately reflect the primary focus of the study. A more appropriate title would be: "The Structure of the Basal Body Inner Junctions Revealed by Electron Cryo-Tomography."

- While the resolution achieved is below 1nm, it may not suffice to unambiguously identify proteins. The authors propose certain assignments based on existing knowledge of ciliary components, but other proteins with similar structures may also fit within the basal body. The current presentation may imply certainty regarding the placement of FAP52 within the inner junction A-B density. I recommend adopting a nomenclature such as "FAP52-like" to better reflect the speculative nature of these assignments. Notably, FAP52/WDR16 could be a paralog of POC16/WDR90, which also shares a domain with FAP20.

- The fit of structures within the cryo-electron tomography (cryo-ET) maps is not always clearly represented in the figures due to the complexity of the densities. It would be beneficial to isolate regions surrounding the fits for better visualization. Supplementary videos, like Video 5, could enhance understanding.

- The observed change in FAP52-like internal density at the proximal end is intriguing; however, the low resolution of the 3D map raises questions about the authors' confidence in this identification. Here again, it may be another protein; the authors should be more speculative in their attributions.

- The authors suggest several MIPs (FAP52, FAP106, IJ34, FAP45, and FAP210) localizing to the inner junction. Could immunofluorescence localization validate this model?

- Figure 4E: Regarding the localization observed via fluorescence microscopy, is it possible to achieve a higher resolution? The data appears to show two centriin dots but only one GFP-Poc39 dot. Is it possible to see whether GFP-Poc39 localizes to microtubule walls using super-resolution? Could they test whether GFP-Poc39 localizes to the probasal body?

-

Minor Comments:

- In Figure S1, indicating which regions correspond to each extracted volume would greatly improve clarity.

- Figure 4: What does the pink dot in the right-hand panel represent?

- The analysis of POC1 depletion leading to replacement by FAP20 and PACRG is robust. This phenomenon resembles the

localization of transition zone elements when specific basal body proteins are depleted in *Chlamydomonas*. Discussing this potential general replacement/substitution mechanism would enrich the discussion, particularly referencing examples such as the delta tubulin mutant (doi/10.1091/mbc.E02-11-0755).

Referee #3:

"Inner junctions" refer to the lumen-facing connections between tubules in the multi-tubule microtubules of cilia. Cryo-EM studies have recently revealed the identities of microtubule inner proteins (MIPs) that bind the IJ of axonemal doublet microtubules (DMTs). However, much less is known about the IJs of the triplet microtubules (TMTs) of the basal body (BB) due to the difficulties of purifying BBs for structural studies.

In this manuscript, Li and colleagues apply cryo-ET and subtomogram averaging to mature BB-axoneme complexes purified from *Tetrahymena thermophila* cilia to obtain subnanometer resolution reconstructions of TMT IJs. The authors subdivide BBs into three regions (proximal, core, and distal) and discover longitudinal variation in the MIPs bound to the IJs. The IJ of the proximal region is bound by POC1 (previously revealed by the same group), FAP52, FAP106, and ladders of unknown identity. FAP52 and FAP106 are also found in axonemal DMTs, potentially consistent with a model where spatial patterning of the axoneme is determined by organization of MIPs in the BB (Ma et al, 2019). The core region of the BB features even more axonemal MIPs (FAP45, FAP210 and IJ34) plus a BB-specific leucine-rich repeat protein identified as Poc39. Localization of Poc39 to the basal body is experimentally validated through colocalization with Centrin, a known basal body protein.

Finally, the authors perform a detailed analysis of TMT IJs in a *T. thermophila* poc1Δ mutant. They identify surprising heterogeneity: some IJs fail to close, some close weakly, while others adopt a structure consistent with the IJ of axonemal DMTs (including FAP20 and PACRG, which are usually excluded from the BB).

This paper advances our understanding of the architecture and spatial patterning of BB TMTs. However, the analysis of the IJs is incomplete and the modest resolutions (although impressive for subtomogram averaging) prevent the identification of some of the novel MIPs and confidence in some of the conclusions. Overall, I recommend publication once the following issues are addressed:

Major comments

1. The manuscript lacks sufficient information about the B-C junctions and any information for the IJs of the distal BB. Why these crucial areas have been neglected is not revealed. Their inclusion in the present study is needed to provide a full description of the IJs in the cilium, as expected from the title.
2. Insufficient information is provided to demonstrate that the correct periodicities have been determined. Does the proximal region have 16 nm periodicity, or is this periodicity simply used as a tool to improve the resolution of the proteins that do follow 16-nm periodicity? This is an important issue as applying an incorrect periodicity during processing can result in features being missed.
3. Further information is required to demonstrate that the assigned proteins agree with the density. For example, the fit of FAP106 to the proximal map (Fig. 2A, B) is unconvincing without additional figures/videos.
4. The authors need to demonstrate that the protein identified as FAP52 in the proximal IJ is not another protein with two beta-propeller domains. An analysis of the *Tetrahymena* genome indicates that it expresses several similar proteins. A different protein might explain why the protein adopts a different position from FAP52 in axonemal DMTs.
5. The beginning of the results section needs to be expanded to give more background about what is being imaged and how. The current manuscript relies on prior knowledge of Li et al, 2022 and Ruehle et al, 2024. Also, it was unclear if the data in this manuscript are identical to Ruehle et al, 2024? If so, why do the numbers of particles used to reconstruct the proximal A/B IJ differ (5834 vs. 12290) differ despite identical resolutions (9.82 Å)?
6. The paper reports models and code (RANSAC) that need to be made available to the community prior to publication. ModelArchive may be a suitable location for the pseudoatomic models.
7. Additional labeling is necessary to improve interpretability of the figures. Each map should be labeled with where it came from (proximal, core, axonemal), the resolution and the periodicity applied. A key should be provided for Fig. 5G.

Minor comments

1. The title and/or abstract must mention the species as it is not yet known if the reported architectures are conserved between species
2. Evolutionary distribution of Poc39 should be discussed. Is this protein specific to *Tetrahymena*?
3. Fig. 1 could be improved by adding all the WT maps described in the manuscript, adding the length of the axoneme, and labeling the transition zone.
4. In Fig. 2 there is an inconsistency between the colors used in the schematic and in the models (i.e. green is used for the ladder in the schematic but for tubulin in panel B).
5. Fig. 5E and the section describing CCDC81b/BMIP1 are confusing. Consider simply showing Fig. S5C next to an atomic model of the DMT IJ that includes CCDC81b and BMIP1.
6. Providing cryo-ET resolutions to two decimal places is unnecessarily precise

7. Fig. 6A combines data from the WT and *poc1* Δ mutant. However, this fails to clearly describe the phenotype of the *poc1* Δ mutant where 3 distinct classes were observed. Consider using panel A for a summary of the WT and panel B for a summary of the mutant.

Referee #1:

Li and colleagues reconstructed 3D structure of the basal body (BB) from *Tetrahymena* at ~9Å resolution. With this pseudo-atomic resolution and referring to known atomic structure of *Chlamydomonas axoneme* (by the Brown group), they identified molecules at the inner junction of triplet microtubules (BB TMT) and built models of three flagellar regions - proximal and core parts of BB (170nm and 300nm length, respectively), as well as the axoneme. They assigned one internal density of TMT to FAP52, a known microtubule internal protein (MIP) of the axoneme. This FAP52 is located differently between the proximal and central core parts of BB. While the location at the central core is similar to that in the axoneme, FAP52 is shifted towards an adjacent microtubule protofilament. Another highlighted protein is Poc1. In their reconstructions, Poc1 is located at the inner junction between A- and B-tubules and between B- and C-tubules. Poc1 knockout induces loss of other MIPs, according to their subtomogram classification, and causes distortion of BB.

While this reviewer appreciates their novel findings, which will encourage the cilia/centriole community for further functional and structural studies on these proteins, a few clarifications should be done. After the authors' addressing the following points, this reviewer will support publication in the EMBO Journal.

Major points:

Despite the well fitted FAP52 atomic model to their cryo-ET density map, at this resolution, possibility that this density is from another protein, which has similar structure as FAP52, cannot be excluded. Even if experimental proof, such as genetic tagging, is not realistic, the authors could still make their assignment more convincing by AlphaFold-multimer to demonstrate the two types (proximal BB and axoneme) of tubulin-FAP52 interaction, or by proving the absence of similar folding prediction from the *Tetrahymena* genome or BB proteomics.

This is a major point made by reviewers. We thank the reviewers for pointing it out and we apologize that we did not address this issue enough in our previous version.

Tetrahymena has three previously reported paralog genes for FAP52 (Kubo, et al. 2023), corresponding to UniProt IDs, Q22ZH2, I7MJ23, and Q24C92. In addition to these three paralogs, our BLAST search has found a fourth FAP52-like protein in *Tetrahymena* (UniProt I7MLQ3, 747 residues), which is larger than the other three (~630 aa).

The AlphaFold2 predicted models for these proteins have two β -propeller domains; each domain has seven WD40 repeats. They fit into our BB map from the proximal region reasonably well. Currently, at subnanometric resolution, we could not definitely identify which one of the paralog proteins is in the proximal region of the basal body A/B inner junction.

Following reviewer #1's suggestion, to further investigate the interaction between FAP52 and MT wall, we conducted a set of AlphaFold-multimer predictions by AlphaFold3. We used two sets of input in two independent runs; In the first run, the input contained two copies of α -tubulin, two copies of β -tubulin, and one FAP52a. In the second run, we used four α -tubulin, four β -tubulin, and a FAP52a as input. The results are shown in the figure below. Despite that the resulting multimer models correctly predicted the tubulin interactions and built MT lattice; in both cases, the FAP52a positions are not consistent with either of the FAP52 positions observed in our experimental map. Specifically, in the (2Tua+2Tub+FAP52a) predicted model (the most left in the figure below), the two β -propeller domains of FAP52a longitudinally span an α -tubulin (in green) on the luminal side of an MT protofilament; in the predicted (4Tua+4Tub+FAP52a) model (2nd from the left), FAP52a binds on the exterior surface of the MT wall. These output models are different from the FAP52 position observed in the experimental maps, as shown in Figs 2B and 4B (both models are shown on the right side of the figure below).

These analyses indicate that FAP52a, by itself, may not have a strong preferred binding site or motif for the luminal side of the MT lattice.

It is plausible that other unidentified MIPs in the B-tubule, for example, the A-B ladder structure unique to the proximal region, could influence the binding site of FAP52a or other FAP52 isoforms in the A/B inner junctions.

Meanwhile, recent high-resolution single particle cryoEM structures on the axonemal DMT have implicated that FAP52 might bind to acetylated K40 in the α -tubulins at the A/B inner junction (Ma et al., 2019; Khalifa et al. 2020). Interestingly, U-ExM has revealed the difference in acetylation state between procentriole and mature centriole (Guichard, et al. 2023). It shows that while the mature centriole is fully acetylated, the procentriole, which becomes the proximal region of the mature centriole later on, lacks the acetylation signal in its early assembly stage. Therefore, it is possible that the difference in the acetylation stage might

regulate the position of FAP52 at the inner junction. Future investigation is needed to test all these hypotheses.

Overall, given that FAP52 has multiple paralog proteins with similar 3D structures, many factors could influence the FAP52-like protein's binding to the MT, our current structures at resolution $\sim 9 \text{ \AA}$ cannot unambiguously confirm it is FAP52a in these regions of BB, following the reviewer's suggestion, we have modified the text and used the term "a FAP52-like protein in the proximal region that binds to pf B9"

They classified subtomograms and showed two ways of FAP52 location/orientation gradually altering from the proximal to the central parts of BB and also two classes. A similar plot is shown for presence and absence of inner scaffold. How are distributions of these multi-structures in BB? It should be possible to show presentation, similarly to Fig.S5G to examine how this transition occurs.

We thank the reviewer's suggestion. We conducted the suggested experiment and have mapped and plotted the location of the 4 classes mentioned in Fig. 3. The new figure depicts the distribution of these 4 classes by using 8 basal bodies as examples. Each dotted line in the figure represents a triplet MT, spanning 450 nm from the proximal to the core region (an arrow indicates the polarity). The 450 nm longitudinal span is divided into 3 segments, each 150 nm long.

In this new figure (as Figure 3F), the green dots represent subtomograms in Class 1 in Figure 3A, an inner junction structure in the proximal region. The red dots represent subtomograms in Class 2 in Figures 3A and S3A, where a FAP52-like protein shifts from pf B9 to the canonical position between pf B9 and B10. The blue dots represent Class 6 in Figures 3E and S3C, where the A-C linker recedes, and the inner scaffold emerges. The yellow dots represent the rest five classes in Figure S3C, where the A-C linker is absent, and the inner scaffold is fully assembled in the TMT. These are the TMT in the core region. These mappings and plots depict our finding that a series of sequential events take place in the inner junctions prior to and during the transition from the proximal to the central region (Page 10).

Meanwhile, we have changed the color in Figs 3B and 3D. Now, the curves in the histograms in these figures are consistent with the colors used in Fig 3F.

These illustrations will provide a direct visualization of the distribution and transition of the above-mentioned structures in the BB.

In their previous work (Ruehle et al. 2024), three regions (proximal, central and distal) parts of BB were compared. In this manuscript, the 100nm length distal region is not mentioned and they compare proximal and central BB regions and the axoneme. It will be fair to show structure of the third BB region. Or is the distal region not involved in BB by their preparation?

Thanks for bringing this point up. Indeed, the distal structure is very interesting. We recently reported a structure from the distal 100 nm of the BB (Ruehle *et al* 2024, EMD-42778). We were not able to obtain a higher resolution structure in this region. This is likely due to substantial structural heterogeneity in the distal region. Several factors might contribute to this heterogeneity: 1) the triplet MT makes a transition to the doublet MT in the distal region, indicating a dynamic change in the composition and structure. 2) The structure in this region might be labile. Our current BB purification procedure might introduce additional perturbations that might limit achievable resolution. 3) Limited number of particles for average due to the relatively short stretch in this region (~100 nm). Obtaining a higher resolution structure from the distal region remains one of the future directions.

We have now clarified this in the *method* section by stating “Due to extensive heterogeneity and a limited number of particles for averaging, the B-C inner junction and the distal region of the basal body are excluded from this study.”

Minor points:

p.19, line 8: Please refer a literature that proved that the proximal region precedes the core region in the BB generation.

Now, we are citing the original study by Richard Allen (1969) showing the stepwise formation of the Tetrahymena basal body from the proximal region by electron microscopy.

Allen, R.D. 1969. THE MORPHOGENESIS OF BASAL BODIES AND ACCESSORY STRUCTURES OF THE CORTEX OF THE CILIATED PROTOZOAN TETRAHYMENA PYRIFORMIS. *J Cell Biology*. 40:716–733. doi:10.1083/jcb.40.3.716.

p.20, line 3: In addition to structural alteration, does BB undergo angle change? Twist is observed during transition from BB to the axoneme. Does this twist occur within the BB?

Our study described in this manuscript, is mainly focusing on the inner junctions. We have not analyzed the change of microtubule lattice throughout the longitudinal length of the basal body or any other factors that might cause the twist within the BB or during the transition from BB to the axoneme. It requires expanding the structural study to other parts of the triplet MT, which deserves full investigation in the future.

Fig.2B: Difference between upper and lower panels should be explained.

The lower panel in Fig 2B intends to show the binding of FAP52 at different protofilament locations in the proximal and core regions, without showing averaged maps in the background. We realize this may cause confusion. Now we have removed this lower panel.

p.34, line6: RANSAC must be explained. Is there a literature?

We have prepared a public version of the RANSAC program and documentation with a detailed description of the procedure implemented, how to use the program, examples, and reference literature. This material is available at the website: <https://tiny.cc/ransac>

In the manuscript, we have added a link to that website so that the readership interested in the method can easily access it.

Table 1: Are the number of particles mentioned here the number finally used for averaging? If so, how many particles did the authors start classification/averaging? How were the resolution calculated - which mask did they use to calculate FSC "at the A-B inner junction", for example?

Yes, the particle numbers mentioned in Table 1 are the numbers used in the final averages. For the WT BB structure, we started with 156 tomograms. Initially, ~40,000 particles were extracted from the core region of BB with substantial overlapping between neighboring TMT segments. We sorted out the MIPs' periodicity by 3D refinement and classification in the program Relion. This was followed by shifting and putting MIPs with different periodicity into their correct registers. The steps reduced the number of particles to 34,006 for the 16-nm repeat structure (EMD-46440) and 4664 for the 48-nm repeat structure (EMD-46439) from the core region. Similarly, from the proximal region, we initially extracted 19,656 particles from 156 BB

tomograms. The final 16-nm repeat structure (EMD-46437) is the average of 12290 particles. Now we have added this information in the "Online Methods" section.

The resolution was assessed by the Fourier Shell Correlation (FSC) between two independent half datasets, using a cutoff value of 0.143, commonly known as the "gold standard" method. Care was taken to ensure the two-half datasets were independent without any overlap. When calculating the FSC, a customized generous soft-edge mask was imposed onto each half-map. The artifacts that might be introduced by the mask were minimized by the phase-randomization procedure, as implemented in Relion (Chen *et al.* *Ultramicroscopy* 35:24–35, 2013. doi:10.1016/j.ultramic.2013.06.004). The FSC curves presented in Fig S1 are the result of these processes. The masks, the two half maps, and validation reports are available from the EMDB deposits.

Now we have added this information in the "Online Methods" section of the manuscript.

Referee #2:

The article by Sam Li and colleagues presents a structural analysis of the inner junction of the triplet microtubule within the basal body and cilium. This inner junction refers to the closure of the B or C microtubule with the A or B microtubule, respectively. Utilizing sub-tomogram averaging, which achieves sub-nanometric resolution across three distinct regions - the proximal region of the basal body, the central core, and the axoneme - the authors conducted a comparative analysis that reveals structural variations along the triplet microtubule of the basal body across these regions. The quality of the work is high, offering valuable insights into the field of basal body and cilium research. Below are some comments for the authors to consider before publication:

- The title does not accurately reflect the primary focus of the study. A more appropriate title would be: "The Structure of the Basal Body Inner Junctions Revealed by Electron Cryo-Tomography."

We thank and agree with the reviewer. We have changed the title as suggested.

- While the resolution achieved is below 1nm, it may not suffice to unambiguously identify proteins. The authors propose certain assignments based on existing knowledge of ciliary components, but other proteins with similar structures may also fit within the basal body. The current presentation may imply certainty regarding the placement of FAP52 within the inner junction A-B density. I recommend adopting a nomenclature such as "FAP52-like" to better reflect the speculative nature of these assignments. Notably, FAP52/WDR16 could be a paralog of POC16/WDR90, which also shares a domain with FAP20.

We agree with the reviewer. Our response about FAP52 is presented above in response to Reviewer 1's first "major concern" on Page 1.

- The fit of structures within the cryo-electron tomography (cryo-ET) maps is not always clearly represented in the figures due to the complexity of the densities. It would be beneficial to isolate regions surrounding the fits for better visualization. Supplementary movies, like Movie 5, could enhance understanding.

We thank the reviewer's suggestion. To improve the visual clarity of the fitting, we have substantially revamped the figures with clear labels. In addition, we made 4 new figures and 3 new movies. We hope they will improve the visualization of the corresponding results.

These 4 new figures are:

1. Figure 2A is now a surface-rendered average map from the proximal region of the basal body where the MIPs are highlighted. This is a better presentation than the previous one, which only shows the atomic models fitting into the map. The previous Figure 2A has been moved to Figure S2A.
2. Figure S2A shows the fitting of atomic models of FAP106, FAP52, and Poc1 into the basal body proximal region map. It includes 3 new inset panels, showing the quality of fitting each of these 3 MIPs in an enlarged view.
3. Figure 4A is a surface-rendered average map from the core region of the basal body. It provides an improved visualization of the MIPs compared to the previous one.
4. Similar to Figure S2A, a new Figure S4A provides 6 panels showing the fitting of the models in the core region into the averaged map, providing improved visualization of modeling the MIPs in their experimental density.

The 3 new movies are:

1. Movie 1 is related to Figure 2A. It shows the surface-rendered 16-nm repeat map (at 9.8 Å resolution) from the proximal region, where the MIP FAP106, FAP52, and Poc1 are highlighted in different colors.
2. Movie 4 is related to Figure 4A. It shows the surface-rendered 48-nm repeat map (at 9.3 Å resolution) from the core region, where all the MIPs are highlighted in different colors.
3. Movie 5 is related to Figures 4B and 4C. It shows the surface-rendered 16-nm repeat map (at 8.3 Å resolution) from the core region, where all the MIPs are highlighted in different colors.

- The observed change in FAP52-like internal density at the proximal end is intriguing; however, the low resolution of the 3D map raises questions about the authors' confidence in this identification. Here again, it may be another protein; the authors should be more speculative in their attributions.

This is related to the second point made by the reviewer #1. We agree with the reviewer's opinion. We have adopted the term "FAP52-like protein". It could be FAP52 or other FAP52

paralog proteins, which we could not unambiguously resolve in our current resolution. We also clarify this point in the discussion section to emphasize this uncertainty.

- The authors suggest several MIPs (FAP52, FAP106, IJ34, FAP45, and FAP210) localizing to the inner junction. Could immunofluorescence localization validate this model?

We appreciate the reviewer's suggestion. Unlike the LRR-motif MIP that is unique to the basal body, FAP52, FAP106, IJ34, FAP45, and FAP210 are shared by both basal body and flagellar axoneme. Localizing these MIPs in the basal body requires resolving the basal body and the axoneme. Either super-resolution light microscopy or expansion light microscopy has to be applied. This is non-trivial and deserves full investigation in the future. Meanwhile, as mentioned above, we are now providing a set of figures presenting details of the fitting. These new figures, based on our current knowledge of the ciliary components, show the assignments are confident.

- Figure 4E: Regarding the localization observed via fluorescence microscopy, is it possible to achieve a higher resolution? The data appears to show two centrin dots but only one GFP-Poc39 dot. Is it possible to see whether GFP-Poc39 localizes to microtubule walls using super-resolution? Could they test whether GFP-Poc39 localizes to the probasal body?

-

We thank the reviewer's suggestion. We agree that light microscopy methods with higher resolution, such as super-resolution and expansion microscopy techniques, are necessary to further confirm the LRR-motif MIP's identity in the core region of BB.

Given that our current result is based on conventional immune-fluorescence microscopy, we are not able to draw a definitive conclusion on the protein's identity or its precise localization in the BB. More experiments are needed. Meanwhile, experiments such as expansion microscopy will take a significant amount of effort and time, therefore, we have decided to remove the immune-fluorescence result previously presented in Figure 4E. Instead, we propose that protein (UniProt Q22N53) is a potential candidate for the LRR-motif protein observed in our structure and we emphasize in the main text (page 12) that *"the confirmation of its identity and its function in BB assembly and maintenance have to await future studies."*

Regarding the localization of LRR-motif MIP to the MT wall or probasal body by using super-resolution light microscopy, first, we have not found any leucine-rich repeat domain in the same location in our averaged inner junction map from the proximal region of the BB. Besides binding to the MT wall, the LRR-motif MIP makes potential interactions with IJ34, which is only present in the core region of the basal body. Furthermore, at the same location in the proximal region, we found the A-B inner ladder, a ladder-like structure linking pf B10 to A13. Overall, we believe that this LRR-motif MIP is unique to the central core region of BB.

Minor Comments:

- In Figure S1, indicating which regions correspond to each extracted volume would greatly improve clarity.

We have put a label on each panel of Fig S1 to denote the name of the map and its location in the basal body.

- Figure 4: What does the pink dot in the right-hand panel represent?

The pink dot in Figure 4 represents an unknown protein that connects Poc1 to the inner scaffold in the central core region of the basal body. We reported this structure in our recent publication (Ruehle, et al 2024). We realize this may cause confusion and have removed this in the revision.

- The analysis of POC1 depletion leading to replacement by FAP20 and PACRG is robust. This phenomenon resembles the localization of transition zone elements when specific basal body proteins are depleted in *Chlamydomonas*. Discussing this potential general replacement/substitution mechanism would enrich the discussion, particularly referencing examples such as the delta tubulin mutant (doi/10.1091/mbc.E02-11-0755).

We thank the reviewer for reminding us of this study. Indeed, this might share a similar mechanism for the basal body assembly. We have now cited this work in the discussion.

Referee #3:

"Inner junctions" refer to the lumen-facing connections between tubules in the multi-tubule microtubules of cilia. Cryo-EM studies have recently revealed the identities of microtubule inner proteins (MIPs) that bind the IJ of axonemal doublet microtubules (DMTs). However, much less is known about the IJs of the triplet microtubules (TMTs) of the basal body (BB) due to the difficulties of purifying BBs for structural studies.

In this manuscript, Li and colleagues apply cryo-ET and subtomogram averaging to mature BB-axoneme complexes purified from *Tetrahymena thermophila* cilia to obtain subnanometer resolution reconstructions of TMT IJs. The authors subdivide BBs into three regions (proximal, core, and distal) and discover longitudinal variation in the MIPs bound to the IJs. The IJ of the proximal region is bound by POC1 (previously revealed by the same group), FAP52, FAP106, and ladders of unknown identity. FAP52 and FAP106 are also found in axonemal DMTs, potentially consistent with a model where spatial patterning of the axoneme is determined by organization of MIPs in the BB (Ma et al, 2019). The core region of the BB features even more axonemal MIPs (FAP45, FAP210 and IJ34) plus a BB-specific leucine-rich repeat protein identified as Poc39. Localization of Poc39 to the basal body is experimentally validated through colocalization with Centrin, a known basal body protein.

Finally, the authors perform a detailed analysis of TMT IJs in a *T. thermophila* poc1Δ mutant.

They identify surprising heterogeneity: some IJs fail to close, some close weakly, while others adopt a structure consistent with the IJ of axonemal DMTs (including FAP20 and PACRG, which are usually excluded from the BB).

This paper advances our understanding of the architecture and spatial patterning of BB TMTs. However, the analysis of the IJs is incomplete and the modest resolutions (although impressive for subtomogram averaging) prevent the identification of some of the novel MIPs and confidence in some of the conclusions. Overall, I recommend publication once the following issues are addressed:

Major comments

1. The manuscript lacks sufficient information about the B-C junctions and any information for the IJs of the distal BB. Why these crucial areas have been neglected is not revealed. Their inclusion in the present study is needed to provide a full description of the IJs in the cilium, as expected from the title.

We have recently reported a structure of the B-C junction from the basal body proximal region in (Ruehle, *et al*, 2024, available in EMD-42781). The resolution of the structure is relatively low, at 10 Å. Here, we reported a structure from the same region, with slightly improved resolution (9.8 Å, EMD-46438). Despite additional tomograms and a larger dataset, we could not achieve higher resolution in order to gain further insight into the molecular detail of the B-C junction. For these reasons, we did not report these regions in our current manuscript.

Similarly, we have recently presented a structure from the distal 100 nm of the BB, reported in (Ruehle et al 2024, available in EMD-42778). However, we were not able to obtain a higher resolution structure in this region. We speculate that this is mainly due to structural heterogeneity in the distal region. Pursuing a higher resolution structure of the distal region remains one of our goals in the future.

In the revision, we updated this information in the Online Methods section (page 39, line 10) by stating *“Due to extensive heterogeneity and limited number of particles for averaging, the B-C inner junction and the distal region of the basal body are excluded from this study.”*

2. Insufficient information is provided to demonstrate that the correct periodicities have been determined. Does the proximal region have 16 nm periodicity, or is this periodicity simply used as a tool to improve the resolution of the proteins that do follow 16-nm periodicity? This is an important issue as applying an incorrect periodicity during processing can result in features being missed.

In the inner junctions, we have found the proximal region of the basal body has a 16 nm periodicity, while both the BB core region and the axoneme of the flagellum have an overall 48 nm periodicity. We used focused 3D classification extensively to find the correct periodicity for each MIP and to put the MIPs in their correct registers.

For the proximal region, on page 7 line 9, we state *“Like the axoneme, the FAP52-like protein and FAP106 have a longitudinal periodicity of 16 nm (Figure 2A)”*

For the core region, on page 11, line 8, we state *“These include FAP52, FAP106, IJ34, FAP45, and FAP210. These MIPs are in the same locations as the axoneme DMT, with the same longitudinal periodicity.”*

Meanwhile, we provide a detailed description in the “Online methods” section. On page 38, line 12, we state *“Customized soft-edge binary masks were used during classification to limit the analysis to the structure of interest. This was critical for determining the correct periodicity of the MIPs and identifying structural defects or heterogeneity in the structure. Once the correct periodicity was found, these out-of-register subtomograms were re-centered and re-extracted. This was followed by combining all subtomograms for the next round of refinement.”*

3. Further information is required to demonstrate that the assigned proteins agree with the density. For example, the fit of FAP106 to the proximal map (Fig. 2A, B) is unconvincing without additional figures/movies.

We have now provided a new set of figures and movies to improve the visualization and to show the fitting of the MIP models and their assignments.

For a detailed description of the new figures and movies, please refer to our response to Reviewer #2 's third major comment.

4. The authors need to demonstrate that the protein identified as FAP52 in the proximal IJ is not another protein with two beta-propeller domains. An analysis of the *Tetrahymena* genome indicates that it expresses several similar proteins. A different protein might explain why the protein adopts a different position from FAP52 in axonemal DMTs.

We agree with the reviewer. Given that our current low resolution and multiple paralog FAP52 genes in *Tetrahymena* genome, we have now adopted the term “FAP52-like” in the proximal inner junction. We also discuss this limitation in more detail in the revision. Please refer to our response about FAP52 above in response to Reviewer #1's first “major concern”.

5. The beginning of the results section needs to be expanded to give more background about what is being imaged and how. The current manuscript relies on prior knowledge of Li et al, 2022 and Ruehle et al, 2024. Also, it was unclear if the data in this manuscript are identical to Ruehle et al, 2024? If so, why do the numbers of particles used to reconstruct the proximal A/B IJ differ (5834 vs. 12290) differ despite identical resolutions (9.82 Å)?

We described the sample preparation, their imaging condition, and image processing in detail in the “Online method” section.

Regarding the question on the dataset size reported previously in Ruehle *et al* 2024 and reported in this study, we appreciate the review’s careful reading of the manuscript. Our previous work described in (Ruehle *et al* 2024), was mainly focused on Poc1. In the work, by comparing the basal body structure of the WT to the *poc1*Δ, we identified Poc1 as a BB inner junction component. Meanwhile, we described the effect of Poc1 deletion on the structure of the basal body. The result described in this work is an extension of our previous work. Here, instead of focusing solely on Poc1, we expanded our study to the inner junctions in three BB and flagellar regions. Meanwhile, we use larger datasets. For example, in Ruehle *et al*, 2024, we used 5834 particles from 63 BB tomograms, resulting in a 9.8 Å averaged structure at the A/B inner junction in the proximal region. In our current study, we used 12290 particles from 156 BB tomograms. Despite the increased dataset size, the resolutions remained the same or slightly better compared to the previously published results. However, the quality of structures improves. This is indicated by comparing their Fourier Shell Correlation (FSC) curves, as shown below on the three structures from the A/B and B/C inner junctions from the proximal region as well as the A/B inner junction in the core region, in 16-nm repeat. Overall, in all three cases, the new FSC curves (in green color) are “healthier” with higher FSC values than the previous ones, indicating improved average map quality across a broad frequency range.

B/C inner junction in the proximal region of basal body (16-nm repeat)

A/B inner junction in the core region of basal body (16-nm repeat)

6. The paper reports models and code (RANSAC) that need to be made available to the community prior to publication. ModelArchive may be a suitable location for the pseudoatomic models.

Regarding the RANSAC code, we have made it available to the community through the website: <https://tiny.cc/ransac> . In the manuscript, we have added a link to that website for the readership interested in the program and its documentation.

We built the pseudo-atomic models either by using previous high-resolution cryoEM (SPA) structures (PDB_8G2Z, PDB_8V3I), or by searching the AlphaFold Database or by using the AlphaFold3 online server (<https://alphafoldserver.com/welcome>). Due to the low-resolution nature of our maps, only rigid-body fitting was used (ChimeraX command “fitmap”) when fitting the models into our low-resolution density maps, without applying any additional refinement steps to modify the models. Though ModelArchive provides a site where any pseudo-atomic can be deposited, either from experimental or theoretical data, unlike PDB, it lacks a rigorous validation process to ensure the quality of the deposition. We opt for a traditional option and will be happy to provide the models to the public upon request. We have added this information in the “Online Methods” section.

7. Additional labeling is necessary to improve interpretability of the figures. Each map should be labeled with where it came from (proximal, core, axonemal), the resolution and the periodicity applied. A key should be provided for Fig. 5G.

We have made additional figures and movies. We put additional labels in Fig S1, including the name, the periodicity, and the resolution for each map.

Minor comments

1. The title and/or abstract must mention the species as it is not yet known if the reported architectures are conserved between species

We have changed the title to “The Structure of the Basal Body Inner Junctions from *Tetrahymena* Revealed by Electron Cryo-Tomography”, and have added “*Tetrahymena*” to the abstract

2. Evolutionary distribution of Poc39 should be discussed. Is this protein specific to *Tetrahymena*?

Our BLAST search for UniProt Q22N53 does not find any obvious homolog in other organisms except in a number of ciliates, such as *Paramecium tetraurelia* (UniProt A0EAA0), *Stentor coeruleus* (UniProt A0A1R2C581), *Pseudocohnilembus persalinus* (UniProt A0A0V0R346), indicating this protein might be unique to the phylum *Ciliophora*. Consistent with this notion, IJ34, which interacts with Poc39, is also a ciliate-specific MIP.

However, LRR is a common motif in BB proteome. Despite their sequence divergence, their structures and functions might be conserved.

Since this is the first time basal body A/B inner junction structures are reported, it is unclear whether this LRR-motif protein is unique to *Tetrahymena*, or a similar MIP might be present in other organisms. Hence, its evolution conservation remains unclear and it has to await structure studies from other species.

3. Fig. 1 could be improved by adding all the WT maps described in the manuscript, adding the length of the axoneme, and labeling the transition zone.

We thank the reviewer's suggestion. However, Fig 1 is meant to be an "introduction figure" showing a stereotypical ultrastructure of BB and axoneme in *Tetrahymena*. Presenting all WT maps in Fig 1 might be overwhelming.

4. In Fig. 2 there is an inconsistency between the colors used in the schematic and in the models (i.e. green is used for the ladder in the schematic but for tubulin in panel B).

We thank the reviewer for pointing this out. Now we have changed the color to light pink for the ladder structures in the inner junctions and green for α -tubulin.

5. Fig. 5E and the section describing CCDC81b/BMIP1 are confusing. Consider simply showing Fig. S5C next to an atomic model of the DMT IJ that includes CCDC81b and BMIP1.

We thank the reviewer's suggestion. Fig 5E shows that in addition to FAP20/PACRG, other axonemal MIPs, such as CCDC81b/BMIP1, are incorporated into the mutant BB as well. The figure intends to emphasize that the mutant BB structure has been partially "morphed" into the axoneme-like structure. To make this clear, now we have modified the figure. The updated Figure 5E shows the atomic models for CCDC81b/BMIP1 fit into the mutant density map.

In addition, we have updated Fig S5C by adding a figure showing a side-by-side comparison of the *poc1Δ* mutant structure to the axoneme where CCDC81b/BMIP1 and other MIPs in the inner junction are highlighted.

6. Providing cryo-ET resolutions to two decimal places is unnecessarily precise

We agree with the reviewer. We have changed and updated the reported resolutions to a single decimal.

7. Fig. 6A combines data from the WT and *poc1Δ* mutant. However, this fails to clearly describe the phenotype of the *poc1Δ* mutant where 3 distinct classes were observed. Consider using panel A for a summary of the WT and panel B for a summary of the mutant.

We thank the reviewer's suggestion. We have moved panel B in Fig. 6A to Fig. 6B.

Dr. Sam Li
University of California, San Francisco
Genentech Hall, MC2242
600, 16th Street
San Francisco, California 94143

10th Feb 2025

Re: EMBOJ-2024-119050R
The Structure of Basal Body Inner Junctions from Tetrahymena Revealed by Electron Cryo-Tomography

Dear Dr. Li,

Thank you for submitting your revised manuscript, as well as for making the final text modifications in response to the re-reviews copied below. I am pleased to inform you that given the referees' overall satisfaction with the revisions, we have now accepted the study for publication in The EMBO Journal.

With kind regards,

Hartmut Vodermaier

Referee #2:

I thank the authors for their thoughtful responses to my questions. The manuscript has undergone significant improvements, and the conclusions are now well-aligned with the findings and resolution of this work. I congratulate the authors for their rigorous approach and valuable contributions to the field. This work not only meets high scientific standards but also sets a new benchmark in the field. I strongly recommend it for publication.

Referee #3:

The authors have made significant improvements to the paper, and the revised title now better reflects the content of the manuscript. The new figures and movies effectively show the map quality and the fit of the models.

Overall, I am satisfied with the responses to my previous comments, although I still feel the Results section begins abruptly. Providing more context would be helpful to the reader, even if this information is available in the Methods section. I also feel that model deposition, even in a database that doesn't provide the same validation as the PDB, is preferable to no deposition at all.

The language around the FAP52-like protein could also be refined. In the rebuttal, the authors mention there are four possible FAP52-like proteins; this information should be included in the main text as well. Statements suggesting that the FAP52-like protein "shifts its location" should also be revised to reflect a model where the different densities indicate a transition from one FAP52-like protein to another.